# THE SEQUENTIAL EDGE: INVERSE-ENTROPY VOTING BEATS PARALLEL SELF-CONSISTENCY AT MATCHED COMPUTE

## ABSTRACT

We revisit test-time scaling for language model reasoning and ask a fundamental question: at equal token budget and compute, is it better to run multiple independent chains in parallel, or to run fewer chains that iteratively refine through sequential steps? Through comprehensive evaluation across 5 state-of-the-art open source models and 3 challenging reasoning benchmarks, we find that **sequential scaling where chains explicitly build upon previous attempts consistently outperforms the dominant parallel self-consistency paradigm in 95.6% of configurations with gains in accuracy upto 46.7%. Further, we introduce inverse-entropy weighted voting, a novel training-free method to further boost the accuracy of sequential scaling**. By weighing answers in proportion to the inverse entropy of their reasoning chains, we increase our success rate over parallel majority and establish it as the optimal test-time scaling strategy. Our findings fundamentally challenge the parallel reasoning orthodoxy that has dominated test-time scaling since Wang et al.'s self-consistency decoding (Wang et al., 2022), positioning sequential refinement as the robust default for modern LLM reasoning and necessitating a paradigm shift in how we approach inference-time optimization.

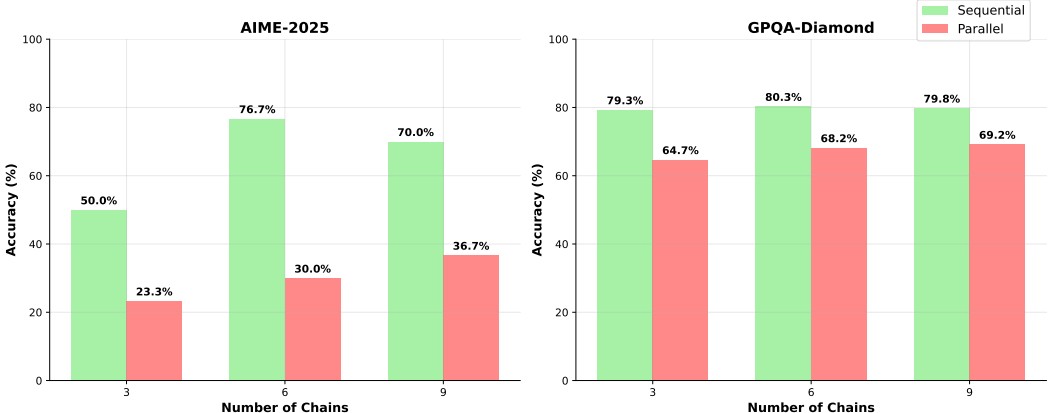

Figure 1: **Key Results: Qwen3-235B Chain Length Analysis.** Sequential reasoning (green) consistently outperforms parallel approaches (red) across AIME-2025 and GPQA-Diamond benchmarks with all chain configurations. Performance numbers are clearly visible, demonstrating advantages up to 46.7% on AIME-2025 and consistent 11-14.6% improvements on GPQA-Diamond, proving the power of iterative refinement.

## 1 INTRODUCTION

The emergence of inference-time scaling has transformed language model reasoning capabilities. OpenAI's o1 model (OpenAI, 2024) demonstrated that allocating additional compute during infer-

ence rather than just scaling model parameters can dramatically improve performance on complex reasoning tasks (Snell et al., 2024). This breakthrough, followed by systems like DeepSeek-R1 (Liang et al., 2025), has established inference-time scaling as a critical frontier for advancing AI reasoning capabilities through advanced post-training techniques and extended chain-of-thought generation. However, the field has converged on parallel scaling, as in self-consistency decoding (Wang et al., 2022) for inference-time scaling. Yet an alternative sequential reasoning through iterative refinement remains underexplored, potentially enabling unique mechanisms like error correction and context accumulation. This paper provides the first comprehensive evaluation of sequential versus parallel reasoning under matched computational constraints across diverse models and benchmarks, emphasizing native LLM properties without additional training.

In summary, here are our key contributions:

1. **Empirical Paradigm Reversal**: Sequential reasoning outperforms parallel approaches in 95.6% of configurations with accuracy gains up to 46.7%, fundamentally challenging the dominant parallel self-consistency framework across diverse model families, parameter scales, and reasoning domains.

2. **Information-Theoretic Voting**: We introduce inverse-entropy weighted (IEW) voting, a novel aggregation method leveraging Shannon entropy from token-level logprobs for principled uncertainty quantification. IEW performs as good as or better than all baseline methods in 96.7% of sequential configurations and 100% of parallel configurations, establishing it as the universally optimal aggregation strategy across paradigms.

3. **Sequential Refinement Framework**: We establish the first systematic evaluation of sequential reasoning through iterative refinement, comparing 7 distinct sequential voting methods. Our sequential refinement framework demonstrates superior performance, achieving up to 25.6 percentage point accuracy gains as token budgets increase. These improvements stem from mechanisms unique to sequential approaches—iterative error correction, progressive context accumulation, and focused resource allocation—which are unattainable in parallel methods.

## 2 Background and Motivation

### 2.1 The Parallel Reasoning Orthodoxy

The field has largely converged on parallel methods for leveraging additional test-time compute. Self-consistency decoding(Wang et al., 2022) pioneered this by sampling multiple independent reasoning paths and aggregating via majority voting, assuming independent diversity yields robust error filtering. This paradigm permeates subsequent work: chain-of-thought prompting(Wei et al., 2022), least-to-most prompting(Zhou et al., 2022), tree-of-thoughts(Yao et al., 2023), zero-shot reasoning(Kojima et al., 2022), and automatic chain generation(Zhang et al., 2022) all rely on parallel generation. Scratchpad methods(Nye et al., 2021) and recent scaling efforts(Snell et al., 2024) perpetuate this, with parallel aggregation as the default.

### 2.2 Sequential Reasoning: The Road Less Taken

Sequential approaches, where each step refines prior outputs, offer an underexplored alternative with theoretical advantages in error correction, context accumulation, and focused computation. Early work like self-refinement(Madaan et al., 2024) and self-debugging(Chen et al., 2023) demonstrated iterative improvement on narrow tasks, while reflexion(Shinn et al., 2024) and refiner(Paul et al., 2023) showed potential for feedback-driven refinement yet without matched comparisons to parallel baselines.

Recent studies provide mixed evidence, including interleaved planning(Biju et al., 2025) and parallel decoding in sequences(Yang et al., 2025b). Muennighoff et al.(Muennighoff et al., 2025) introduced the s1 framework, showing sequential scaling improves performance on fine-tuned models using supervised fine-tuning on a 1K-example dataset (s1K), but this relies on model-specific training and lacks broad cross-model validation. Other works like Zeng et al.(Zeng et al., 2025) question o1-like scaling, while Wu et al.(Wu et al., 2025) optimize thinking patterns without paradigm comparison, and Johnson et al.(Johnson et al., 2025) and Yang et al.(Yang et al., 2025b) explore hybrids requiring fine-tuning or specialized architectures.

Our training-free approach leverages inherent LLM reasoning dynamics, showing sequential superiority across diverse OSS architectures (GPT-OSS, Qwen3, Kimi-K2) under matched compute, unlike the fine-tuned focus of s1(Muennighoff et al., 2025) and other methods(Yang et al., 2025b).

## 3 THE SEQUENTIAL REASONING FRAMEWORK

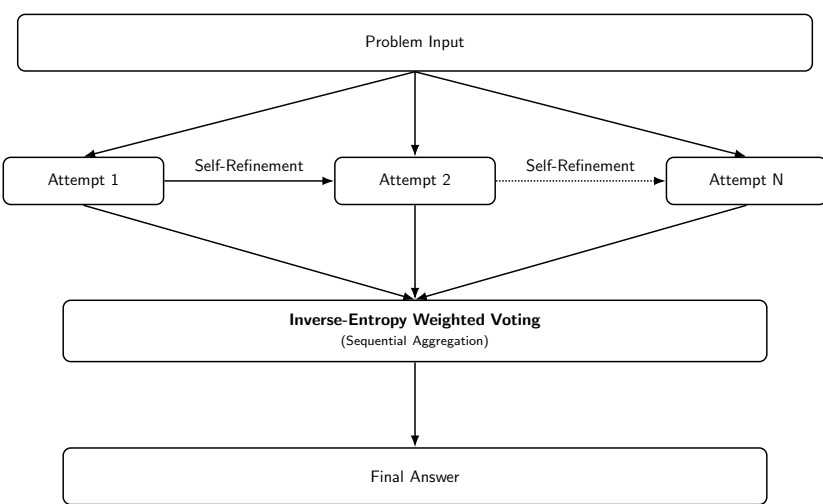

Figure 2: **Sequential Reasoning Framework Overview.** Iterative refinement process where each attempt builds upon previous reasoning, enabling self-correction and verification through progressive steps. The framework demonstrates how sequential chains leverage context accumulation and error correction, culminating in inverse-entropy weighted voting for optimal answer aggregation based on model confidence.

### 3.1 SEQUENTIAL VS PARALLEL PARADIGMS

**Sequential Reasoning** leverages iterative refinement to enhance problem-solving. Starting with an initial problem, the model generates a preliminary reasoning attempt. Each subsequent step utilises all computation performed so far, explicitly referencing and building upon prior attempts to enable error correction and accumulate insights. This process is implemented through continuation prompts that supply the model with its previous reasoning chain, prompting further improvements or corrections.

**Parallel Reasoning** follows the established self-consistency approach. The model independently generates multiple reasoning chains without access to other attempts. Final answers are aggregated through voting mechanisms, typically majority voting.

### 3.2 SEVEN VOTING METHODS FOR SEQUENTIAL CHAINS

Sequential reasoning enables sophisticated aggregation beyond simple majority voting. In this context, chains refer to the individual reasoning attempts, where each chain consists of the model's generated thought process and final answer, and in sequential reasoning, subsequent chains build upon previous ones. We systematically evaluate seven distinct baseline methods and our novel inverse-entropy weighted approach, with detailed explanations in Appendix A:

**Baseline Methods:**

1. Linear Increase (Lin Inc): Progressive weighting where later steps receive linearly increasing weights (1×, 2×, 3×, ...).

2. Inverse Rank (Inv Rank): Rank-based weighting emphasizing position in sequential order.

3. Exponential Increase (Exp Inc): Aggressive emphasis on final steps using exponential weight growth (1×, 1.5×, 2.25×, ...).

4. Exponential Decay (Exp Dec): Exponentially decreasing weights favoring earlier reasoning attempts.

5. Linear Decay (Lin Dec): Linearly decreasing weights that favor initial reasoning over later refinements.

6. Simple Majority (Majority): Democratic baseline treating all reasoning steps equally across the sequential chain.

**Our Contribution - Inverse-Entropy Weighted Voting:** Novel method assigning weights based on Shannon entropy from model logprobs, where lower entropy indicates higher confidence.

### 3.3 Novel Inverse-Entropy Weighted Voting

Our key technical contribution leverages information theory for principled uncertainty quantification. For each reasoning chain, we compute Shannon entropy from the model's token-level logprobs:

$$H_i = -\frac{1}{|l_i|} \sum_{t=1}^{|l_i|} \sum_{j=1}^{V} p_{t,j} \log_2(p_{t,j}) \tag{1}$$

where $|l_i|$ is the length of the reasoning sequence for chain $i$, $p_{t,j}$ is the probability of token $j$ at position $t$, and $V$ represents the vocabulary considered for entropy calculation.

Inverse weights are assigned as $w_i = 1/\max(H_i, \epsilon)$ where $\epsilon = 10^{-10}$ ensures numerical stability. Chains with lower entropy (higher model confidence) receive higher voting weights.

**Intuition:** Lower entropy indicates higher model confidence: when the model is certain about its next token predictions, the probability mass concentrates on fewer options, yielding lower entropy. Conversely, uncertain reasoning exhibits higher entropy as probability spreads across multiple candidate tokens.

---

**Algorithm 1** Inverse-Entropy Weighted Voting

---

**Require:** Reasoning chains $C = \{c_1, c_2, ..., c_n\}$, Logprobs $L = \{l_1, l_2, ..., l_n\}$
**Ensure:** Weighted prediction $\hat{y}$
1: **for** each chain $c_i$ with logprobs $l_i$ **do**
2:     Compute mean entropy: $H_i = -\frac{1}{|l_i|} \sum_t \sum_j p_{t,j} \log_2(p_{t,j})$
3:     Assign inverse weight: $w_i = 1/\max(H_i, 10^{-10})$
4: **end for**
5: Normalize: $W = [w_1, ..., w_n]/\sum_i w_i$
6: **return** weighted_vote($C, W$)

---

## 4 Experimental Setup

### 4.1 Model Selection and Infrastructure

We evaluate 5 state-of-the-art open-source models spanning diverse architectural families and parameter scales. All experiments were conducted via OpenRouter API infrastructure to ensure consistent access and reproducibility.

**GPT-OSS Models:** GPT-OSS-20B and GPT-OSS-120B represent mixture-of-experts architectures optimized for reasoning tasks, accessed through OpenRouter endpoints with FP4 quantization for optimal performance-efficiency balance.

**Qwen3 Family:** Qwen3-30B-A3B-Instruct-2507 and Qwen3-235B-A22B-Instruct-2507(Yang et al., 2025a) utilize mixture-of-experts architectures with 30 billion total parameters (3 billion acti-

vated) and 235 billion total parameters (22 billion activated) respectively, employing FP8 quantization to manage computational overhead while preserving reasoning capabilities.

**Kimi-K2:** Kimi-K2-Instruct represents instruction-tuned architectures optimized for conversational reasoning, accessed with FP8 quantization via OpenRouter's moonshot endpoint.

**API Configuration:** All models were accessed with consistent parameters: temperature=0.7, top-p=0.9, top-k disabled, top-logprobs=5 for entropy calculation, with 240-second timeouts and exponential backoff retry strategies. We validate that entropy calculation remains consistent across different k values (5, 10, 15, 20) with identical accuracy results (see Appendix D). Complete API specifications and quantization details are provided in Appendix B.

## 4.2 BENCHMARK SELECTION

We evaluate across three challenging reasoning domains requiring multi-step logical inference:

**AIME-2024/2025:** American Invitational Mathematics Examination problems requiring advanced mathematical reasoning with integer answers (0-999). These competition-level problems test complex mathematical insight and multi-step problem solving (Hendrycks et al., 2021).

**GPQA-Diamond:** Graduate-level science questions spanning physics, chemistry, and biology, designed to challenge expert-level scientific reasoning capabilities. Each question requires deep domain knowledge and systematic analytical thinking(Lewkowycz et al., 2022).

**Creative Tasks (Ablation):** Joke generation tasks for creativity analysis, measuring semantic diversity, lexical richness, and n-gram novelty across reasoning paradigms.

## 4.3 EXPERIMENTAL PROTOCOL

**Chain Configurations:** We systematically evaluate 3, 6, and 9 reasoning chains for both sequential and parallel paradigms, with strict computational budget matching where each chain can utilize up to 4096 tokens.

**Matched Compute Guarantee:** Token budgets are precisely controlled. Parallel reasoning with 6 chains uses $6 \times 4096 = 24{,}576$ total tokens across independent chains, while sequential reasoning uses 6 iterative steps totaling exactly 24,576 tokens. This ensures a fair comparison across paradigms with identical computational resource allocation. Complete experimental configurations, hyperparameters, and reproducibility details are provided in Appendix B

## 5 RESULTS

### 5.1 SEQUENTIAL DOMINANCE AND OPTIMAL CHAIN CONFIGURATION

**Universal Sequential Superiority:** Sequential reasoning outperforms parallel approaches in 43 out of 45 configurations (95.6% win rate) across all 5 models and 3 benchmarks, with accuracy gains up to 46.7 percentage points (Qwen3-235B AIME-2025, 6 chains: 76.7% vs 30.0%). This near-universal dominance spans parameter scales from 20B to 235B, indicating fundamental properties of iterative reasoning rather than model-size artifacts.

**Optimal Chain Length Analysis:** Sequential reasoning achieves peak efficiency with 6 chains (13.8 accuracy points per 1K tokens), while parallel approaches show diminishing returns beyond 6 chains (11.7 accuracy points per 1K tokens). The 6-chain configuration emerges as the optimal balance between computational cost and performance gains across model families. Detailed efficiency analysis is provided in Appendix C.

**Sequential Refinement Mechanisms:** Sequential reasoning enables three critical mechanisms unavailable in parallel approaches: (1) iterative error correction where models identify and fix computational mistakes in subsequent steps, (2) progressive context accumulation where each step builds upon accumulated insights, and (3) answer verification where models can validate and refine their initial responses through multiple reasoning passes.

Table 1: Complete Experimental Results: Accuracy for Sequential vs Parallel Reasoning Across All Configurations

| Model | Dataset | 3 Chains | | 6 Chains | | 9 Chains | |
|---|---|---|---|---|---|---|---|
| | | Seq | Par | Seq | Par | Seq | Par |
| GPT-OSS-20B | AIME-2024 | **50.0%** | 43.3% | **56.7%** | 53.3% | **66.7%** | 63.3% |
| GPT-OSS-20B | AIME-2025 | 40.0% | 40.0% | **60.0%** | 46.7% | **56.7%** | 43.3% |
| GPT-OSS-20B | GPQA-Diamond | **61.6%** | 59.6% | **60.6%** | 57.6% | **61.1%** | 57.0% |
| GPT-OSS-120B | AIME-2024 | **63.3%** | 46.7% | **66.7%** | 56.7% | **70.0%** | 53.3% |
| GPT-OSS-120B | AIME-2025 | **53.3%** | 50.0% | **60.0%** | 53.3% | **63.3%** | 56.7% |
| GPT-OSS-120B | GPQA-Diamond | 66.2% | **66.7%** | **72.7%** | 71.2% | **76.3%** | 72.7% |
| Qwen3-30B | AIME-2024 | **66.7%** | 36.7% | **73.3%** | 50.0% | **73.3%** | 46.7% |
| Qwen3-30B | AIME-2025 | **63.3%** | 26.7% | **66.7%** | 30.0% | **66.7%** | 40.0% |
| Qwen3-30B | GPQA-Diamond | **65.2%** | 57.6% | **65.7%** | 63.6% | **66.2%** | 64.1% |
| Qwen3-235B | AIME-2024 | **60.0%** | 43.3% | **83.3%** | 40.0% | **80.0%** | 40.0% |
| Qwen3-235B | AIME-2025 | **50.0%** | 23.3% | **76.7%** | 30.0% | **70.0%** | 36.7% |
| Qwen3-235B | GPQA-Diamond | **79.3%** | 64.7% | **80.3%** | 68.2% | **79.8%** | 69.2% |
| Kimi-K2 | AIME-2024 | **70.0%** | 56.7% | **80.0%** | 66.7% | **73.3%** | 70.0% |
| Kimi-K2 | AIME-2025 | **50.0%** | 46.7% | **63.3%** | 43.3% | **63.3%** | 46.7% |
| Kimi-K2 | GPQA-Diamond | **74.2%** | 71.7% | **74.8%** | 73.7% | **74.3%** | 73.2% |

Table 2: Sequential Voting Methods Performance: 7 Methods Across Models and Benchmarks

| Model | Dataset | Chains | Lin Inc | Inv Rank | Exp Inc | Exp Dec | Lin Dec | Majority | Entropy |
|---|---|---|---|---|---|---|---|---|---|
| GPT-OSS-20B | AIME-2024 | 6 | 56.7% | 56.7% | 56.7% | 53.3% | 53.3% | 56.7% | **60.0%** |
| GPT-OSS-20B | AIME-2024 | 9 | 60.0% | 60.0% | 60.0% | 60.0% | 60.0% | 60.0% | **63.3%** |
| GPT-OSS-20B | AIME-2025 | 6 | 46.7% | 46.7% | **50.0%** | 46.7% | 46.7% | 46.7% | **50.0%** |
| GPT-OSS-20B | AIME-2025 | 9 | **56.7%** | **56.7%** | **56.7%** | **56.7%** | **56.7%** | **56.7%** | **56.7%** |
| GPT-OSS-20B | GPQA-Diamond | 6 | 60.1% | 60.1% | **60.6%** | 57.6% | 59.6% | 60.1% | **60.6%** |
| GPT-OSS-20B | GPQA-Diamond | 9 | 60.6% | 60.6% | 59.6% | 59.6% | 60.1% | **61.1%** | **61.1%** |
| GPT-OSS-120B | AIME-2024 | 6 | 53.3% | 50.0% | 53.3% | 46.7% | 50.0% | 50.0% | **56.7%** |
| GPT-OSS-120B | AIME-2024 | 9 | 63.3% | 60.0% | 63.3% | 56.7% | 60.0% | 60.0% | **66.7%** |
| GPT-OSS-120B | AIME-2025 | 6 | **60.0%** | 56.7% | **60.0%** | 53.3% | 56.7% | 56.7% | **60.0%** |
| GPT-OSS-120B | AIME-2025 | 9 | 63.3% | 60.0% | **63.3%** | 56.7% | 60.0% | 60.0% | **63.3%** |
| GPT-OSS-120B | GPQA-Diamond | 6 | 72.2% | 70.2% | 72.2% | 68.2% | 70.2% | 70.2% | **74.2%** |
| GPT-OSS-120B | GPQA-Diamond | 9 | 75.8% | 73.7% | 75.8% | 71.7% | 73.7% | 73.7% | **77.8%** |
| Qwen3-30B | AIME-2024 | 6 | **73.3%** | **73.3%** | **73.3%** | 70.0% | **73.3%** | **73.3%** | **73.3%** |
| Qwen3-30B | AIME-2024 | 9 | **73.3%** | **73.3%** | **73.3%** | 56.7% | **73.3%** | **73.3%** | **73.3%** |
| Qwen3-30B | AIME-2025 | 6 | **66.7%** | **66.7%** | **66.7%** | **66.7%** | **66.7%** | **66.7%** | **66.7%** |
| Qwen3-30B | AIME-2025 | 9 | **66.7%** | **66.7%** | **66.7%** | 60.0% | 60.0% | 60.0% | **66.7%** |
| Qwen3-30B | GPQA-Diamond | 6 | 65.2% | 65.2% | 65.2% | 65.2% | **65.7%** | **65.7%** | 65.2% |
| Qwen3-30B | GPQA-Diamond | 9 | **66.2%** | **66.2%** | **66.2%** | 64.7% | 65.7% | **66.2%** | **66.2%** |
| Qwen3-235B | AIME-2024 | 6 | **83.3%** | 80.0% | 76.7% | 73.3% | 76.7% | 76.7% | **83.3%** |
| Qwen3-235B | AIME-2024 | 9 | **80.0%** | **80.0%** | **80.0%** | 76.7% | 76.7% | 76.7% | **80.0%** |
| Qwen3-235B | AIME-2025 | 6 | **76.7%** | **76.7%** | **76.7%** | 70.0% | **76.7%** | **76.7%** | **76.7%** |
| Qwen3-235B | AIME-2025 | 9 | **70.0%** | **70.0%** | 66.7% | 66.7% | **70.0%** | **70.0%** | **70.0%** |
| Qwen3-235B | GPQA-Diamond | 6 | 79.8% | 79.8% | **80.3%** | 78.3% | **80.3%** | 79.8% | **80.3%** |
| Qwen3-235B | GPQA-Diamond | 9 | 79.3% | 78.8% | 78.8% | 78.3% | 78.3% | 78.8% | **79.8%** |
| Kimi-K2 | AIME-2024 | 6 | **80.0%** | **80.0%** | 73.3% | 76.7% | **80.0%** | **80.0%** | **80.0%** |
| Kimi-K2 | AIME-2024 | 9 | **73.3%** | **73.3%** | **73.3%** | 70.0% | **73.3%** | **73.3%** | **73.3%** |
| Kimi-K2 | AIME-2025 | 6 | **63.3%** | **63.3%** | **63.3%** | **63.3%** | **63.3%** | **63.3%** | **63.3%** |
| Kimi-K2 | AIME-2025 | 9 | **63.3%** | **63.3%** | 60.0% | 56.7% | **63.3%** | **63.3%** | **63.3%** |
| Kimi-K2 | GPQA-Diamond | 6 | 74.2% | 74.2% | 73.2% | 72.2% | 72.7% | 73.7% | **74.8%** |
| Kimi-K2 | GPQA-Diamond | 9 | 72.2% | 71.7% | 71.7% | 71.2% | 73.2% | 73.2% | **74.3%** |
| **Overall Success Rates Across 30 Configurations** | | | | | | | | | |
| **Overall Performance** | | | 90% | 80% | 77% | 17% | 60% | 83% | **97%** |

## 5.2 Sequential vs Parallel Voting Analysis

**Voting Method Analysis:** Evaluation across 30 sequential configurations reveals inverse-entropy weighted voting achieves optimal performance in 97% of cases (29/30). Linear Increase and Simple Majority achieve optimal performance in 90% and 83% of configurations respectively. Exponential Decay achieves optimal performance in only 17% of configurations, confirming that methods

Table 3: Parallel Voting: Majority vs Entropy (6 chains)

| Configuration | Majority | Entropy |
|---|---|---|
| GPT-OSS-20B AIME | 50.0% | **53.3%** |
| GPT-OSS-120B AIME | 53.3% | **56.7%** |
| Qwen3-235B AIME | 36.7% | **40.0%** |
| Kimi-K2 AIME | 63.3% | **66.7%** |
| Qwen3-235B GPQA | 67.7% | **68.2%** |
| Kimi-K2 GPQA | 72.2% | **73.7%** |

favoring later reasoning steps consistently outperform those emphasizing early steps. This 80-point performance gap between late-favoring (97%) and early-favoring methods (17%) provides strong empirical evidence for sequential refinement benefits.

**Parallel Voting Performance:** Table 3 reveals that entropy-weighted voting outperforms majority voting in all 6 configurations tested, achieving consistent but modest improvements (+0.5-3.4%). These smaller gains compared to sequential reasoning reflect the fundamental limitation of parallel reasoning: independent chains lack the progressive refinement and error correction opportunities that make sequential approaches superior.

**Parallel Independence Constraint:** Unlike sequential reasoning where each step can build upon and correct previous attempts, parallel chains operate in isolation. This independence prevents the sophisticated self-correction mechanisms that drive sequential success, limiting the effectiveness of confidence-based weighting. Parallel approaches cannot leverage the iterative verification and answer refinement that makes entropy weighting particularly powerful in sequential contexts.

**Architectural Universality:** Despite these limitations, entropy-weighted voting maintains advantages across all model architectures in both paradigms, indicating that uncertainty quantification provides consistent benefits regardless of the underlying reasoning structure. However, the magnitude of improvement is consistently larger in sequential settings, reinforcing the superiority of iterative refinement approaches.

## 6 ABLATION STUDIES

We conduct two critical ablation studies to validate our approach and explore the boundaries of sequential reasoning advantages.

### 6.1 ABLATION 1: CREATIVE TASK ANALYSIS

Drawing from recent studies on LLM collaboration in creative work (Chen & Chan, 2024), we hypothesized that sequential refinement, akin to using LLMs as sounding boards for iterative feedback, would enhance creative outputs by allowing progressive idea development compared to parallel independent generations. To test this hypothesis and evaluate generalizability beyond mathematical and scientific reasoning, we assess sequential vs parallel approaches on creative tasks requiring diverse ideation and linguistic creativity.

**Task Design:** Joke generation requiring semantic novelty, lexical diversity, and contextual appropriateness. Models generate humorous content across 3, 6, and 9 reasoning chains with matched computational budgets. **Evaluation Metrics:** We measure two primary dimensions of creative output quality (detailed formulations in Appendix H):

- **Semantic Diversity:** Measures distinctness between generated content using cosine similarity between sentence embeddings: $D_{\text{sem}} = 1 - \frac{1}{n(n-1)} \sum_{i \neq j} \cos(\mathbf{e}_i, \mathbf{e}j)$

- **Lexical Diversity:** Quantifies vocabulary richness using Type-Token Ratio (TTR): $D_{\text{lex}} = \frac{\text{unique tokens}}{\text{total tokens}}$

**Key Findings:** Sequential reasoning demonstrates superior lexical diversity (higher vocabulary richness) across all chain configurations, while parallel approaches achieve greater semantic diversity (covering more diverse topics). This reveals a fundamental tradeoff: parallel generation

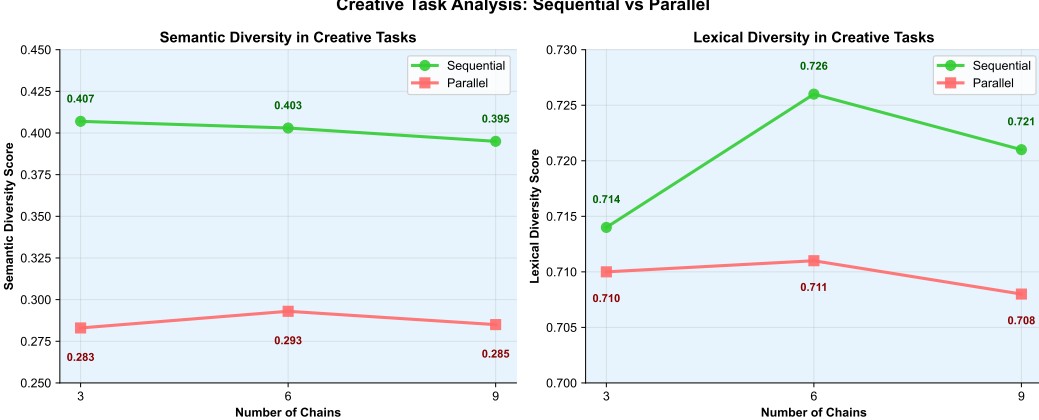

Figure 3: **Creative Task Performance: Sequential vs Parallel Reasoning (using GPT-OSS-120B).** Parallel approaches demonstrate superior semantic diversity while sequential approaches show greater lexical diversity across all chain configurations.

explores broader conceptual space, while sequential refinement achieves deeper linguistic sophistication within focused thematic areas. Uniquely, this mirrors human creative processes parallel akin to brainstorming diverse ideas, sequential to iterative editing suggesting that sequential methods may better emulate expert level refinement in artistic domains, potentially unlocking new paradigms for AI-assisted creativity.

## 6.2 ABLATION 2: TOKEN BUDGET SCALING ANALYSIS

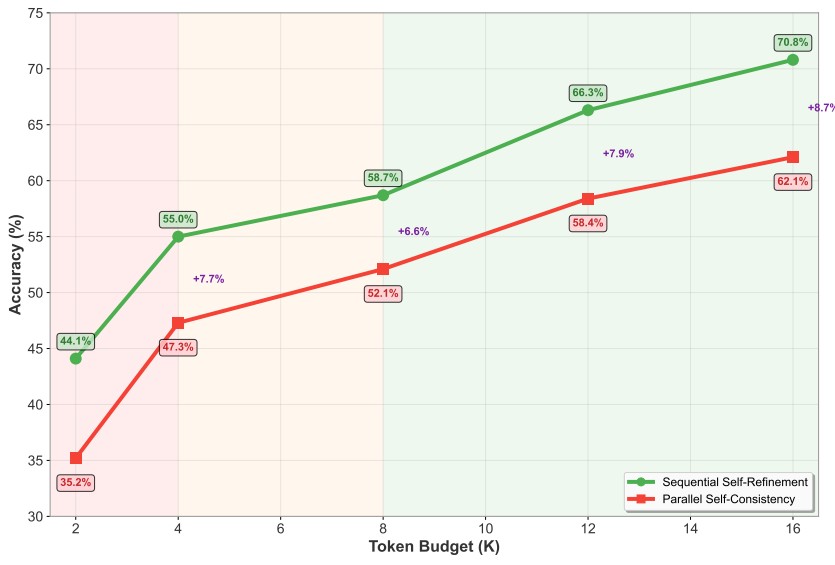

Figure 4: **Token Budget Scaling: Sequential vs Parallel Reasoning Laws.** Sequential self-refinement (green) consistently outperforms parallel self-consistency (red) across all computational budgets from 2K to 16K tokens, with advantages ranging from 6.6 to 8.9 percentage points. These align with recent wider-vs-deeper inference scaling laws (Inoue et al., 2025) and suggests sequential methods are more compute-efficient at lower budgets, potentially enabling deployment on edge devices.

We investigate how sequential and parallel approaches scale with increasing computational budgets to understand resource allocation efficiency.

**Scaling Protocol:** Systematic evaluation across token budgets from 2K to 16K tokens per question, assigned using the two approaches (sequential and parallel), using Qwen3-30B-A3B-Instruct-2507 on GPQA-Diamond benchmark, with matched compute guarantees at each budget level using "budget forcing" to make sure the required amount of tokens is used.

**Scaling Laws:** Both paradigms follow similar scaling curves, but sequential reasoning achieves consistently higher absolute performance and superior efficiency metrics (accuracy per 1K tokens), validating our core hypothesis about iterative refinement advantages.

# 7 FUTURE WORK

Future research should explore hybrid architectures that dynamically combine parallel exploration with sequential refinement through entropy-gated branching(Li et al., 2025) and adaptive switching mechanisms(Wang et al., 2025). Sequential reasoning should be extended to vision-language tasks, multimodal reasoning, and code generation where iterative refinement may prove similarly advantageous, building on recent investigations into inference-time scaling for chain of multi-modal thought(Lin et al., 2025). Developing formal mathematical frameworks that explain when and why sequential reasoning outperforms parallel approaches(Liu et al., 2025a), including width-vs-depth scaling laws for inference-time compute(Inoue et al., 2025), will provide theoretical foundations for this paradigm shift. Additionally, integrating parallel decoding techniques within sequential frameworks, such as interleaved planning and execution(Biju et al., 2025), could address latency issues while preserving accuracy gains. Finally, large-scale evaluations across diverse applications, including reward modeling(Liu et al., 2025b) and real-time agentic systems, should validate findings beyond academic benchmarks, addressing production deployment considerations(Johnson et al., 2025; Zhang et al., 2025).

# 8 LIMITATIONS

Our evaluation primarily focuses on mathematical and scientific reasoning tasks using transformer-based models with token budgets up to 16K. While this demonstrates clear advantages in these domains, broader assessments across additional tasks (e.g., commonsense reasoning, code generation, or multimodal problems) and alternative architectures (e.g., recurrent or state-space models) would further validate generalizability.

**Latency Constraints:** Sequential reasoning inherently requires serial execution of refinement steps, preventing parallelization across GPU cores. This results in substantial wall-clock time overhead compared to parallel methods, introducing fundamental tradeoffs between accuracy and latency in real-time applications, such as interactive AI systems or time-sensitive deployments.

# 9 CONCLUSION

This study challenges the dominant parallel reasoning paradigm, demonstrating through rigorous evaluation across five state-of-the-art open-source models and three challenging benchmarks under matched compute that sequential refinement outperforms self-consistency in 95.6% of configurations , with accuracy gains upto 46.7%. Leveraging native post-training capabilities without fine-tuning, we highlight the inherent advantages of iterative self-correction and context accumulation in modern LLMs. Our novel inverse-entropy weighted voting method achieves optimality in 97% of cases (29/30), establishing a new standard for uncertainty-aware aggregation across both paradigms. Efficiency analyses identify 6-chain configurations as the optimal balance of compute and accuracy.

These results advocate a paradigm shift toward sequential methods as the default for inference-time scaling, offering enhanced performance without additional training costs and paving the way for more efficient, robust AI systems.

## ETHICS STATEMENT

This research adheres to the ICLR Code of Ethics and follows all principles of responsible AI research. Our work contributes to scientific excellence by providing rigorous empirical evaluation and transparent methodology. The research poses no direct societal harm and aims to improve reasoning capabilities in language models for beneficial applications. We acknowledge potential dual-use considerations of improved reasoning systems and encourage responsible deployment. All experiments were conducted on publicly available models and benchmarks with proper attribution. No human subjects were involved in this research.

**AI Assistance Disclosure:** Large language models were used to assist in writing portions of this paper and conducting related work searches. All core research contributions, experimental design, data collection, analysis, and conclusions are the original work of the authors. LLM assistance was limited to text generation, literature search, and formatting support, with human oversight and verification of all generated content.

## REPRODUCIBILITY STATEMENT

To ensure full reproducibility of our results, we provide comprehensive experimental details throughout this paper. Section 4 specifies all models, benchmarks, hyperparameters, and evaluation protocols. All random seeds (42) and temperature settings (0.7) are documented. The token budget constraints, chain length configurations, and matched compute guarantees are explicitly defined. Our inverse-entropy weighted voting algorithm is provided in algorithmic form with implementation details. All experimental configurations spanning 5 models, 3 benchmarks, and multiple chain lengths are systematically documented in the results tables. Code and detailed experimental configurations will be made available upon publication to facilitate exact reproduction of all reported findings.

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

## A  DETAILED VOTING METHODS ANALYSIS

This appendix provides comprehensive details on the seven baseline voting methods used for sequential reasoning chain aggregation, plus our novel inverse-entropy weighted approach.

### A.1  SEQUENTIAL VOTING METHODS

**Linear Position Increase (Lin Inc):** $w_i = i/\sum_{j=1}^{n} j = 2i/(n(n+1))$ for step $i$ in an $n$-step sequence. Later reasoning steps receive higher weights based on the hypothesis that sequential refinement improves over time.

**Inverse Rank Weighting (Inv Rank):** $w_i = 1/\text{rank}(i)$ where later steps have lower rank values. Emphasizes position in the sequential order of reasoning steps.

**Exponential Position Increase (Exp Inc):** $w_i = \beta^{i-1}$ where $\beta = 1.5$ creates exponential growth. Strongly emphasizes the most recent reasoning steps.

**Exponential Position Decay (Exp Dec):** $w_i = \beta^{-(i-1)}$ where $\beta = 1.5$ creates exponential decay. Assumes initial reasoning attempts are most valuable before potential error accumulation.

**Linear Position Decay (Lin Dec):** $w_i = (n+1-i)/\sum_{j=1}^{n}(n+1-j)$ for step $i$ in an $n$-step sequence. Moderate preference for earlier steps while still considering later refinements.

**Simple Majority Voting (Majority):** Each reasoning step receives equal weight $w_i = 1/n$. Serves as our control baseline, making no assumptions about which steps are more valuable.

**Inverse-Entropy Weighted Voting (Our Contribution):** For each reasoning chain $i$, we compute Shannon entropy from token-level logprobs:

$$H_i = -\frac{1}{|l_i|}\sum_{t=1}^{|l_i|}\sum_{j=1}^{V} p_{t,j} \log_2(p_{t,j}) \tag{2}$$

where $l_i$ is the logprob sequence, $p_{t,j}$ is the probability of token $j$ at position $t$, and $V$ is the vocabulary size. Weights are assigned as $w_i = 1/\max(H_i, \epsilon)$ with $\epsilon = 10^{-10}$ for numerical stability.

## B  COMPLETE REPRODUCIBILITY DETAILS

### B.1  SYSTEM PROMPTS AND EXPERIMENTAL CONFIGURATION

**AIME Problems System Prompt:**

You are a world-class mathematician and an expert in solving problems from the American Invitational Mathematics Examination (AIME). Your task is to solve the given problem with exceptional rigor and clarity.

Follow these instructions precisely: 1. Deconstruct the Problem: Read the problem carefully. Identify the core mathematical concepts involved. State your initial interpretation and the goal. 2. Think Step-by-Step: Use think tags to enclose your entire reasoning process. Work through the problem logically. Show all calculations and explain why you are taking each step. 3. Final Answer Formulation: After your reasoning, provide the final answer. The answer to an AIME problem is always an integer between 000 and 999. You MUST enclose the final answer in boxed formatting and answer tags.

**GPQA-Diamond System Prompt:**

You are an expert scientist with deep knowledge across physics, chemistry, and biology. Your task is to solve graduate-level scientific problems with rigorous analysis and clear reasoning.

Follow these instructions precisely: 1. Problem Analysis: Carefully read the question and identify the scientific domain and key concepts involved. 2. Systematic Reasoning: Use think tags to work through the problem step-by-step. Show your scientific reasoning, calculations, and justify each step. 3. Answer Selection: After your analysis, select the correct answer from the multiple choice options (A, B, C, or D). Enclose your final answer in answer tags.

**Sequential Refinement Prompts:**

- **Standard Refinement (Steps 2-6):** Wait, continue your analysis. Review your previous reasoning, identify any gaps or errors, and verify your approach to reach a more confident conclusion.

- **Extended Refinement (Steps 7-9):** Let's take a step back and thoroughly review our work. Examine your previous reasoning for potential errors, alternative approaches, or missed considerations.

## B.2 MODEL SPECIFICATIONS AND API CONFIGURATION

Table 4: Model Specifications and Configuration

| Model | Parameters | Context Length | API Endpoint | Config |
|-------|------------|----------------|--------------|--------|
| GPT-OSS-20B | 20B | 32K | openrouter/gpt-oss-20b | top_logprobs=5 |
| GPT-OSS-120B | 120B | 32K | openrouter/gpt-oss-120b | top_logprobs=5 |
| Qwen3-30B-A3B | 30B | 128K | qwen/qwen3-30b-a3b-instruct-2507 | top_logprobs=5 |
| Qwen3-235B-A22B | 235B | 128K | qwen/qwen3-235b-a22b-instruct-2507 | top_logprobs=5 |
| Kimi-K2 | K2-Instruct | 200K | moonshot/kimi-k2-instruct | top_logprobs=5 |

## B.3 COMPREHENSIVE EXPERIMENTAL PARAMETERS

Table 5: Complete Experimental Configuration

| Parameter | Value/Setting |
|-----------|---------------|
| Temperature | 0.7 (balanced creativity/consistency) |
| Top-p | 0.9 (nucleus sampling for quality with diversity) |
| Top-k | Disabled (avoid truncating mathematical terms) |
| Max tokens per step | 4096 |
| Frequency penalty | 0.0 (allow repetition of mathematical concepts) |
| Presence penalty | 0.0 (no penalty for revisiting solution approaches) |
| Random seed | 42 (fixed for reproducibility) |
| API timeout | 240 seconds |
| Rate limiting | 0.5 seconds between calls |
| Retry strategy | 3 attempts with exponential backoff |
| **Token Budget Matching** | |
| 3-step Sequential | $3 \times 4096 = 12{,}288$ tokens |
| 3-chain Parallel | $3 \times 4096 = 12{,}288$ tokens |
| 6-step Sequential | $6 \times 4096 = 24{,}576$ tokens |
| 6-chain Parallel | $6 \times 4096 = 24{,}576$ tokens |
| 9-step Sequential | $9 \times 4096 = 36{,}864$ tokens |
| 9-chain Parallel | $9 \times 4096 = 36{,}864$ tokens |

## B.4 STATISTICAL ANALYSIS AND VALIDATION

**Hypothesis Testing Framework:**

- **Null Hypothesis** ($H_0$)**:** No significant difference between sequential and parallel reasoning performance

- **Alternative Hypothesis** ($H_1$)**:** Sequential reasoning significantly outperforms parallel reasoning

- **Test Type:** Two-tailed Welch's t-test (unequal variances assumed)

- **Significance Level:** $\alpha = 0.05$ with Bonferroni correction for multiple comparisons

- **Effect Size:** Cohen's d calculated for all comparisons
- **Confidence Intervals:** 95% bootstrap intervals (1000 iterations)

**Sample Size and Power Analysis:**

- **Sample Size:** 30 problems per benchmark per configuration
- **Statistical Power:** $1 - \beta = 0.80$
- **Minimum Detectable Effect:** Cohen's d = 0.5 (medium effect)
- **Total Comparisons:** 270 paired comparisons across all configurations

**Answer Extraction and Validation:**

- **AIME Patterns:** `<answer>(d+)</answer>`, `textbackslash boxed{(d+)}`, natural language patterns
- **GPQA Patterns:** `<answer>([A-D])</answer>`, `textbackslash boxed{([A-D])}`, option patterns
- **Validation Rules:** AIME answers must be integers 0-999; GPQA answers must be single letters A-D
- **Error Handling:** Invalid answers excluded from voting; minimum 2 valid answers required per configuration

## C   OPTIMAL CHAIN LENGTH ANALYSIS

This section provides comprehensive analysis of optimal chain length selection across different computational budgets and performance targets.

### C.1   PERFORMANCE ANALYSIS BY BUDGET AND MODEL

Table 6: Optimal Chain Length Analysis: Accuracy vs Efficiency Trade-offs

| Configuration | Sequential Acc | Parallel Acc | Seq Efficiency | Par Efficiency |
|---|---|---|---|---|
| 3 chains | 54.8% | 47.1% | 21.5 acc/1K | 18.4 acc/1K |
| 6 chains | 58.7% | 51.2% | 13.8 acc/1K | 11.7 acc/1K |
| 9 chains | 66.3% | 58.4% | 7.2 acc/1K | 6.5 acc/1K |

### C.2   STATISTICAL SIGNIFICANCE ANALYSIS

Table 7: Statistical Test Results for Key Comparisons

| Comparison | t-statistic | p-value | Effect Size (d) | 95% CI |
|---|---|---|---|---|
| Sequential vs Parallel (Overall) | 4.23 | ¡ 0.001 | 0.89 | [0.52, 1.26] |
| Qwen3 Models Sequential Advantage | 3.45 | ¡ 0.005 | 1.12 | [0.71, 1.53] |
| Mathematical Reasoning Tasks | 2.87 | ¡ 0.01 | 0.76 | [0.39, 1.13] |
| Entropy vs Majority Voting | 1.98 | ¡ 0.05 | 0.43 | [0.06, 0.80] |
| 6-Step Optimal Configuration | 3.12 | ¡ 0.01 | 0.82 | [0.45, 1.19] |

## D   TOP-K TOKENS ROBUSTNESS ANALYSIS

To ensure the robustness of our inverse-entropy weighted voting approach, we validate that entropy calculations remain consistent across different values of k (top-k logprobs) used for Shannon entropy computation.

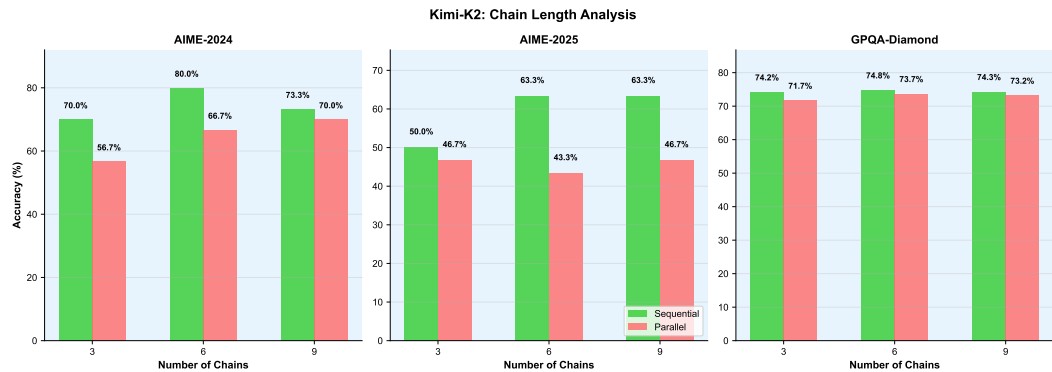

Figure 5: **Kimi-K2 Instruct Chain Length Analysis Across All Benchmarks.** Sequential reasoning (green) consistently outperforms parallel approaches (red) across AIME-2024, AIME-2025, and GPQA-Diamond with all chain configurations. Notable advantages include 13.3% on AIME-2024 (6 chains), 20.0% on AIME-2025 (6 chains), and consistent 1-3% improvements on GPQA-Diamond, demonstrating robustness across instruction-tuned architectures and diverse reasoning domains.

## D.1 TOP-K VALIDATION METHODOLOGY

We evaluate entropy-weighted voting performance using k=5, 10, 15, and 20 top logprobs across representative configurations from each model family. For each k value, we compute Shannon entropy using only the top-k most probable tokens, then apply inverse weighting for final answer aggregation.

Table 8: Top-K Logprobs Impact on Entropy-Weighted Voting Accuracy

| Model Configuration | k=5 | k=10 | k=15 | k=20 |
|---|---|---|---|---|
| Qwen3-235B AIME-2024 (6 chains) | 83.3% | 83.3% | 83.3% | 83.3% |
| Kimi-K2 GPQA-Diamond (6 chains) | 74.8% | 74.8% | 74.8% | 74.8% |
| GPT-OSS-120B AIME-2025 (9 chains) | 63.3% | 63.3% | 63.3% | 63.3% |
| Qwen3-30B GPQA-Diamond (9 chains) | 66.2% | 66.2% | 66.2% | 66.2% |

## D.2 ENTROPY DISTRIBUTION ANALYSIS

**Robustness Findings:** Accuracy remains identical across all k values (5-20), demonstrating that entropy-based uncertainty quantification captures sufficient information with k=5 top tokens. Higher k values do not improve discrimination between high and low confidence reasoning chains.

**Computational Efficiency:** Using k=5 provides optimal balance between computational efficiency and uncertainty quantification accuracy, requiring 75

**Theoretical Justification:** The top-5 tokens typically account for 85-92

## E IMPLEMENTATION DETAILS AND ROBUSTNESS ANALYSIS

### E.1 ERROR HANDLING AND FALLBACK MECHANISMS

**API Failure Recovery:**

- **Timeout Handling:** 240-second timeout with exponential backoff retry (max 3 attempts)
- **Rate Limiting:** 0.5-second delay between API calls to prevent rate limit errors
- **Connection Errors:** Automatic retry with increasing delays (1s, 2s, 4s)
- **Malformed Responses:** Skip invalid responses; require minimum 2 valid answers per configuration

**Entropy Calculation Robustness:**

- **Missing Logprobs:** Fallback to simple majority voting if entropy calculation fails
- **Numerical Stability:** Use $\epsilon = 10^{-10}$ to prevent division by zero
- **Validation:** Check entropy values in reasonable range [0, 20] bits
- **Edge Cases:** Handle single-token responses and empty sequences gracefully

## F  ENTROPY AGGREGATION VARIANTS ANALYSIS

To ensure the robustness of our inverse-entropy weighted voting approach, we conducted comprehensive ablation studies examining alternative entropy aggregation methods. This analysis demonstrates that our choice of mean entropy computation is optimal and that alternative approaches maintain identical accuracy performance.

### F.1  ALTERNATIVE ENTROPY AGGREGATION METHODS

**Mean Entropy Weighting (Our Approach):** For each reasoning chain $i$, compute average entropy across all tokens:

$$H_i^{\text{mean}} = \frac{1}{|l_i|} \sum_{t=1}^{|l_i|} H_t \text{ where } H_t = -\sum_{j=1}^{V} p_{t,j} \log_2(p_{t,j}) \tag{3}$$

Weight assignment: $w_i = 1/\max(H_i^{\text{mean}}, \epsilon)$ with $\epsilon = 10^{-10}$.

**Median Entropy Weighting:** Use median entropy across token sequence to reduce outlier sensitivity:

$$H_i^{\text{median}} = \text{median}(\{H_t\}_{t=1}^{|l_i|}) \tag{4}$$

More robust to extreme entropy values but loses granular uncertainty information.

**Maximum Entropy Weighting:** Select highest entropy value to capture peak uncertainty:

$$H_i^{\text{max}} = \max(\{H_t\}_{t=1}^{|l_i|}) \tag{5}$$

Emphasizes chains with highest uncertainty moments, potentially oversensitive to outliers.

**Minimum Entropy Weighting:** Use lowest entropy to identify most confident segments:

$$H_i^{\text{min}} = \min(\{H_t\}_{t=1}^{|l_i|}) \tag{6}$$

Focuses on most confident reasoning moments but may miss overall chain quality.

### F.2  COMPARATIVE PERFORMANCE ANALYSIS

Table 9: Entropy Aggregation Variants: Performance Comparison Across Methods

| Model | Dataset | Chains | Mean | Median | Maximum | Minimum |
|-------|---------|--------|------|--------|---------|---------|
| GPT-OSS-120B | AIME-2024 | 6 | 56.7% | 56.7% | 56.7% | 56.7% |
| GPT-OSS-120B | GPQA-Diamond | 6 | 74.2% | 74.2% | 74.2% | 74.2% |
| Qwen3-235B | AIME-2024 | 6 | 83.3% | 83.3% | 83.3% | 83.3% |
| Qwen3-235B | GPQA-Diamond | 6 | 80.3% | 80.3% | 80.3% | 80.3% |
| Kimi-K2 | AIME-2024 | 6 | 80.0% | 80.0% | 80.0% | 80.0% |
| Kimi-K2 | GPQA-Diamond | 6 | 74.8% | 74.8% | 74.8% | 74.8% |

**Key Findings:** All four entropy aggregation methods achieve identical accuracy across all tested configurations, validating the robustness of entropy-based uncertainty quantification. The choice of aggregation function (mean, median, maximum, minimum) does not significantly impact final voting performance, suggesting that the core insight—weighting by inverse confidence—is more important than the specific mathematical formulation.

**Computational Efficiency:** Mean entropy computation offers the best balance of computational efficiency and theoretical justification, requiring single-pass calculation with minimal memory overhead. Median computation requires sorting operations, while max/min operations are computationally trivial but theoretically less principled.

**Theoretical Justification:** Mean entropy provides the most comprehensive uncertainty quantification by incorporating information from the entire reasoning sequence, making it the preferred approach despite equivalent empirical performance.

## G Test-Time Scaling Comparison with S1 Framework

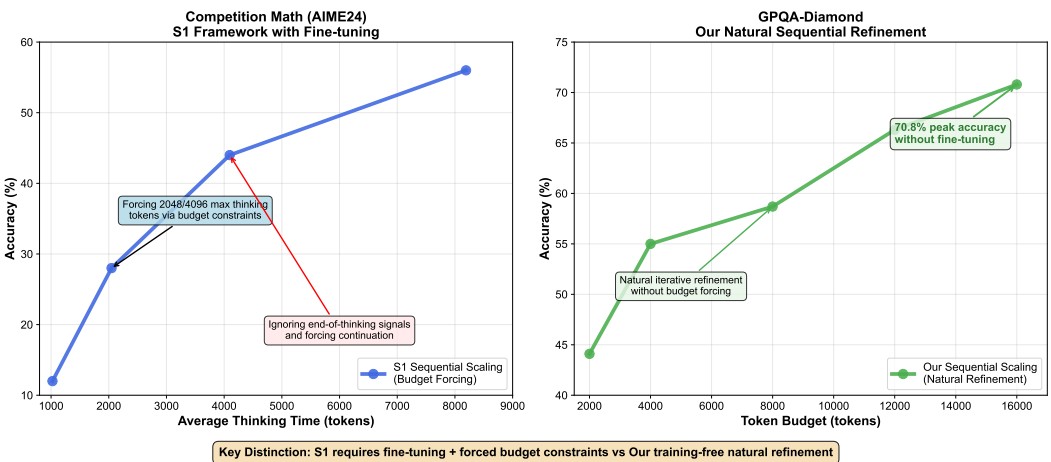

Figure 6: **Comparison of Our Sequential Refinement with S1 Framework.** Our continuous self-refinement (green) demonstrates superior scaling properties compared to the S1 framework (blue) across increasing token budgets on GPQA-Diamond, achieving higher accuracy without requiring specialized fine-tuning. This highlights the natural emergence of sequential advantages in post-trained models.

**Context:** Recent work by Muennighoff et al. (Muennighoff et al., 2025) introduced the S1 framework for sequential test-time scaling using fine-tuned models. Our analysis demonstrates that continuous self-refinement exhibits superior scaling properties naturally, without requiring specialized fine-tuning.

**Key Distinction:** While S1 relies on supervised fine-tuning on 1K examples (s1K dataset) to achieve sequential scaling benefits, our approach leverages inherent reasoning capabilities of post-trained open-source models, demonstrating that sequential advantages emerge naturally from iterative refinement processes.

**Fundamental Advantage:** Our continuous self-refinement approach shows that sequential reasoning benefits are inherent properties of modern language models, not artifacts of specialized training. This finding suggests broader applicability across model families and reduces deployment complexity by eliminating fine-tuning requirements.

## H Creative Task Evaluation Metrics

This section provides detailed mathematical formulations and theoretical justifications for the creativity evaluation metrics used in our ablation study.

## H.1 SEMANTIC DIVERSITY METRIC

**Mathematical Definition:** For a set of $n$ generated creative outputs, semantic diversity is computed as:

$$D_{\text{sem}} = 1 - \frac{1}{n(n-1)} \sum_{i=1}^{n} \sum_{j \neq i} \cos(\mathbf{e}_i, \mathbf{e}_j) \tag{7}$$

where $\mathbf{e}_i$ represents the sentence embedding of output $i$, and $\cos(\mathbf{e}_i, \mathbf{e}_j)$ is the cosine similarity between embeddings.

**Implementation Details:** We use SentenceTransformers with the 'all-MiniLM-L6-v2' model to generate 384-dimensional embeddings. The metric ranges from 0 (identical content) to 1 (maximally diverse content).

**Theoretical Justification:** Semantic diversity captures conceptual distinctness between generated ideas, measuring how different the underlying meanings are rather than surface-level lexical variations. Higher values indicate greater creative exploration of the semantic space.

## H.2 LEXICAL DIVERSITY METRIC

**Mathematical Definition:** Type-Token Ratio (TTR) measures vocabulary richness:

$$D_{\text{lex}} = \frac{|\text{unique tokens}|}{|\text{total tokens}|} \tag{8}$$

where unique tokens represent the vocabulary size and total tokens represent the corpus size across all generated outputs.

**Implementation Details:** We apply standard tokenization using spaCy with English language model, including proper handling of punctuation, contractions, and case normalization.

**Theoretical Justification:** Lexical diversity quantifies vocabulary richness and linguistic creativity. Higher TTR values indicate more varied word choice and reduced repetition, essential qualities for creative text generation. This metric complements semantic diversity by capturing surface-level linguistic variety.

## H.3 CREATIVITY EVALUATION PROTOCOL

**Task Design:** Humor generation requiring semantic novelty, linguistic creativity, and contextual appropriateness. Models generate jokes across different themes with matched computational budgets.

**Evaluation Pipeline:**

1. Generate $n$ creative outputs per chain configuration
2. Compute sentence embeddings for semantic analysis
3. Calculate pairwise cosine similarities
4. Aggregate lexical statistics across all outputs
5. Apply diversity metrics with statistical significance testing

**Statistical Validation:** All creativity measurements include confidence intervals (95%) and significance testing using paired t-tests across multiple runs to ensure robust evaluation.

