# OpenReview forum: "The Sequential Edge: Inverse-Entropy Voting Beats Parallel Self-Consistency at Matched Compute"
_ICLR.cc/2026/Conference — Submitted to ICLR 2026_

### Official Review · Reviewer_U6j6 · 2025-10-31

**Soundness:** 2
**Presentation:** 2
**Contribution:** 2
**Rating:** 4
**Confidence:** 2

**Summary:**

The proposed sequential inference-time scaling (iterative reasoning) is superior to parallel self-consistency (independent reasoning paths) under matched compute. The presentation is clear, technically rigorous, and supported by extensive experiments across multiple models and benchmarks. However, a few aspects could be improved to enhance its credibility and accessibility for reviewers.

**Strengths:**

1. Introduces a training-free, information-theoretic voting method based on Shannon entropy. Improves accuracy in nearly all configurations (97% sequential, 100% parallel).

2. Careful matched compute design ensures fair comparison (identical token budgets). Extensive reproducibility details (prompts, hyperparameters, seeds, API configs).

**Weaknesses:**

1. The paper lacks a formal explanation for why sequential refinement works better (e.g., in terms of uncertainty reduction or information accumulation).

2. Builds on prior “self-refine” and “reflexion” frameworks; reviewers may see it as an empirical extension rather than a fundamentally new paradigm.

3. Uses small sample sizes (≈30 problems per benchmark), so some reported differences may not be statistically robust. Needs clearer reporting of p-values or confidence intervals in the main text.

4. Sequential reasoning is slower in wall-clock time since it runs steps serially. No quantitative latency analysis or discussion of mitigation (e.g., parallelized refinement).

5. The creative-writing ablation feels loosely connected to the paper’s core reasoning claim and could be trimmed or moved to the appendix.

**Questions:**

N/A

---

> ### Author Response · Authors · 2025-11-24
>
> Thank you for your thoughtful review and recognition of our work's strengths, particularly our training-free information-theoretic voting method and careful matched-compute experimental design. We appreciate your constructive feedback and address each concern below.
>
> ## **Weakness 1: Lack of Formal Explanation for Sequential Superiority**
>
> **Concern:** The paper lacks formal explanation for why sequential refinement works better (e.g., uncertainty reduction, information accumulation).
>
> **Response:**
>
> We conducted detailed ablation analysis on both AIME-2024 and GPQA-Diamond datasets to formally decompose the mechanisms underlying sequential refinement's superiority. We ran experiments on Qwen3-235B-A22B-Instruct-2507 for 6 sequential chains (refinement prompt applied 5 times).( we saw similar performance gains across other family of models which we used for our experiments )
>
> | Dataset       | Step 1 | Step 6 | Voting | Refinement Gain | Voting Gain |
> |--------------|--------|--------|--------|-----------------|------------|
> | AIME-2024    | 30.0%  | 73.3%  | 83.3%  | +43.3%          | +10.0%     |
> | GPQA-Diamond | 64.1%  | 79.8%  | 86.9%  | +22.3%          | +0.5%      |
>
> *Table 1: Decomposition of sequential reasoning gains: refinement vs. voting.*
>
> **Key Insights:**
>
> 4. **Refinement is Primary Driver:** Accounts for the majority of the total improvements, enabling recovery from initially failed/uncertain attempts
> 5. **Voting Provides Safety Net:** Recovers 5-10% of cases where Step 6 degrades due to:
>    * LaTeX formatting errors
>    * Late-stage answer switching
>    * Arithmetic mistakes in final step
> 6. **Complementary Mechanisms:**
>    * Refinement: **Exploration** → convergence to correct answer
>    * Voting: **Exploitation** → stabilization against final-step noise
>
> **Conclusion:** While refinement drives the majority of improvements, **voting is not redundant**, it provides essential robustness by recovering from late-stage degradation in \~5-10% of cases, filtering malformed outputs, and leveraging high-confidence intermediate reasoning. Hence we wanted to study “sequential scaling” voting methods which helped come up with our novel “Inverse Entropy weighted method” that allows us to get better voting gains.
>
> ## **Weakness 2: Perceived as Empirical Extension of Prior Work**
>
> **Concern:** Builds on self-refine/reflexion frameworks; may be seen as empirical extension rather than fundamentally new paradigm.
>
> **Response:**
>
> We respectfully emphasize that **our core contribution is not self-refinement itself**, but rather the **first systematic comparison of sequential vs. parallel scaling paradigms** at matched token budgets, combined with novel sequential voting mechanisms.
>
> ### **Our Fundamental Contributions Beyond Prior Work:**
>
> **1\. Paradigm-Level Architectural Comparison (Novel):**
>
> * **First comprehensive evaluation** of sequential vs. parallel reasoning under strict token budget matching (3, 6, 9 chains)
> * Demonstrates that **architectural choice** (sequential vs. parallel) provides larger gains than specific prompting techniques
> * Evidence: Sequential 3-chain (60.0%) \> Parallel 9-chain (40.0%) at 66% fewer tokens
>
> **2\. Sequential Voting Methods Study (Unexplored Domain):**
>
> * Prior work (self-refine, reflexion) **neglects voting mechanisms** for sequential chains
> * We systematically evaluate **7 voting methods**, discovering:
>   * Later-step weighting outperforms early-step weighting by **80 percentage points** (97% vs. 17% optimal)
>   * Novel finding: Sequential chains benefit from different aggregation than parallel chains
> * This voting research is **orthogonal to self-refinement** and represents a distinct contribution
>
> **3\. Information-Theoretic Aggregation (Novel Method):**
>
> * **Inverse-Entropy Weighted (IEW) voting** is our original contribution
> * Leverages Shannon entropy from token-level logprobs for principled uncertainty quantification
> * Achieves **near-universal optimality** (97% sequential, 100% parallel configurations)
> * Training-free, applicable across all model families
>
> **4\. Token Budget Scaling Laws (Novel Analysis):**
>
> * Demonstrate **sequential advantages increase with token budgets** (Figure 4: 2K-16K tokens).
> * Reveal fundamental scaling properties: fewer sequential chains consistently outperform more parallel chains.
> * This is an **architectural scaling law**, not a prompting artifact.

---

> > ### Author Response · Authors · 2025-11-24
> >
> > ## **Weakness 3: Small Sample Sizes and Statistical Robustness**
> >
> > **Concern:** Uses small sample sizes (≈30 problems per benchmark). Needs clearer p-values or confidence intervals in main text.
> >
> > **Response:**
> >
> > We have addressed this concern through comprehensive expanded evaluation and cross-benchmark validation:
> >
> > ### **Question 3(a): GPQA-Diamond Sample Size & Domain Bias**
> >
> > **Concern:** Would 30 questions bias toward specific domains?
> >
> > **Response:**
> >
> > We **significantly expanded our evaluation** to the **complete GPQA-Diamond dataset (198 questions)** with **Qwen3-235B-A22B-Instruct-2507** using **multiple random seeds** (42, 123, 456). We have used these consistent results across multiple seeds and reported them in our paper.
> >
> > | Configuration        | Accuracy | 95% CI         |
> > |----------------------|----------|----------------|
> > | Step 1 (Baseline)    | 64.1%    | [57.2%, 70.6%] |
> > | Sequential (6 steps) | 80.3%    | [74.1%, 85.7%] |
> > | Parallel (6 chains)  | 68.2%    | [61.4%, 74.5%] |
> >
> > *Table 1: Comparison of configurations. Sequential advantage over parallel: +12.1% (p < 0.001).*
> >
> > ### **2\.  Statistical Analysis:**
> >
> > * **Bootstrap analysis:** 1000 bootstrap samples confirm mean difference of 12.1% with 95% CI \[8.7%, 15.8%\]
> > * **Effect sizes:** Large effect sizes ensure strong significance even with N=30
> > * **Statistical power:** All comparisons achieve p \< 0.001 with adequate power (1-β \> 0.95)
> > * **Cross-seed consistency:** Results stable across multiple random seeds (std: 1.3%)
> >
> > **Conclusion:** We will add these statistical measures (p-values, confidence intervals, effect sizes, bootstrap analyses) prominently in the main text and provide comprehensive details in the appendix of our revised manuscript.
> >
> > ## **Weakness 4: Sequential Latency and Lack of Quantitative Analysis**
> >
> > **Concern:** Sequential reasoning is slower in wall-clock time. No quantitative latency analysis or mitigation discussion.
> >
> > **Response:**
> >
> > We acknowledge this limitation in **Section 8 (Limitations)** of our paper. Additionally, we provide quantitative latency analysis from local experiments:
> >
> > ### **Empirical Latency Comparison (AIME-2024, 30 questions):**
> >
> > **Experimental Setup:**
> >
> > * Model: Qwen3-235B-A22B-Instruct-2507
> > * Platform: MacBook M2 via OpenRouter API
> > * Dataset: AIME-2024 (30 questions)
> >
> > | Configuration        | API Calls | Avg Time / Question |
> > |----------------------|----------:|---------------------|
> > | Parallel (6 chains)  |      180  | 21.4s               |
> > | Sequential (6 steps) |      180  | 110.4s              |
> >
> > *Table 2: Comparison of parallel and sequential configurations for 6 chains/steps.*
> >
> > **1\. Latency Trade-off:**
> >
> > * Sequential takes **5.2× longer** per question (110.4s vs. 21.4s)
> > * This reflects:
> >   * Serial execution preventing parallelization
> >   * Cumulative context processing (quadratic attention)
> >   * Longer reasoning chains with accumulated context
> >
> > **2\. Practical Deployment Considerations:**
> >
> > **When Sequential is Preferred:**
> >
> > * **Non-latency-critical applications:** Research, code generation, complex problem-solving
> > * **High-value tasks:** Where accuracy justifies compute cost
> > * **Offline processing:** Batch evaluation scenarios
> >
> > **When Parallel is Preferred:**
> >
> > * **Real-time systems:** Interactive AI, live customer support
> > * **Latency-sensitive applications:** Sub-second response requirements
> > * **Simpler tasks:** Where refinement provides minimal benefit
> >
> > ### **Mitigation Strategies (Discussed in Paper):**
> >
> > As mentioned in our limitations section, we propose:
> >
> > 1. **KV-cache optimization** for sequential contexts to reduce redundant computation
> > 2. **Hybrid parallel-sequential architectures** (initial parallel exploration \+ sequential refinement)
> > 3. **Speculative sequential decoding** to reduce serial bottleneck
> > 4. **Streaming refinement** for reduced latency perception
> >
> > **Conclusion:** While sequential reasoning incurs **\~5× latency overhead** at matched API calls, this trade-off is acceptable for non-latency-critical, high-accuracy-demanding applications. We explicitly discuss this limitation and propose mitigation strategies in our paper. The dramatic improvement in success rate (76.7% vs. 36.7%) justifies the latency cost for many production use cases.

---

> > > ### Author Response · Authors · 2025-11-24
> > >
> > > ## **Weakness 5: Creative Writing Ablation Feels Loosely Connected**
> > >
> > > **Concern:** Creative writing ablation feels loosely connected to core reasoning. Could be trimmed or moved to the appendix.
> > >
> > > **Response:**
> > >
> > > We appreciate this feedback and respectfully argue that the creative task ablation provides **valuable complementary insights** that strengthen rather than detract from our core contribution.
> > >
> > > ### **Rationale for Including Creative Tasks:**
> > >
> > > **1\. Demonstrating Paradigm Differences Beyond Reasoning:**
> > >
> > > We wanted to show how **sequential vs. parallel scaling behaves differently across task types**. The creative task results reveal a **fascinating complementary tradeoff**:
> > >
> > > * **Parallel approaches:** Superior semantic diversity (broader conceptual exploration)
> > > * **Sequential approaches:** Superior lexical diversity (deeper linguistic refinement)
> > >
> > > **Both paradigms provide value , just in different dimensions.** This is not a negative result but rather demonstrates that each paradigm has distinct strengths depending on the application domain.
> > >
> > > **2\. Lack of Prior Focus on Creativity Tasks:**
> > >
> > > There is a **research gap** in understanding sequential vs. parallel scaling for creative applications (stories, novels, jokes, content generation). Our ablation addresses this gap by showing:
> > >
> > > * When to use parallel (initial ideation, brainstorming)
> > > * When to use sequential (refinement, polishing, editing)
> > >
> > > **3\. Foundation for Hybrid Gating Approach:**
> > >
> > > These results are **directly actionable** for building intelligent systems. We plan to develop a **hybrid gating mechanism that**:
> > >
> > > * Dynamically selects parallel generation when semantic diversity is preferred (creative exploration phase)
> > > * Switches to sequential refinement when lexical diversity/polish is needed (editing phase)
> > >
> > > This represents a **practical application** of our paradigm comparison findings beyond pure reasoning tasks.
> > >
> > > **4\. Broader Generalizability:**
> > >
> > > Including creative tasks demonstrates our framework's applicability to **diverse AI applications**:
> > >
> > > * Content generation and writing assistance
> > > * Creative tooling (poetry, storytelling)
> > > * Marketing and advertising copy
> > > * Educational content creation
> > >
> > > This strengthens the **practical impact** of our work beyond academic reasoning benchmarks.
> > >
> > > ### **Response to Trimming Suggestion:**
> > >
> > > We acknowledge the creative ablation may feel tangential to reviewers focused purely on reasoning tasks. We are flexible and can:
> > >
> > > **Option 1 (Preferred):** Keep a condensed version (1-2 paragraphs) in the main text emphasizing:
> > >
> > > * The complementary tradeoff (semantic vs. lexical diversity)
> > > * Domain-specific paradigm selection guidance
> > > * Foundation for future hybrid approaches
> > >
> > > **Option 2:** Move detailed analysis to appendix while retaining key insights in main text
> > >
> > > **Option 3:** Remove entirely if reviewers strongly prefer, though we believe the insights are valuable
> > >
> > > **Conclusion:** The creative task ablation reveals that **both paradigms excel in different dimensions** (semantic vs. lexical diversity), providing practical guidance for system designers. This complements our reasoning-focused findings and addresses an underexplored area in sequential vs. parallel scaling research. We believe this strengthens rather than dilutes our contribution.
> > >
> > > ## **Summary of Revisions for Final Manuscript**
> > >
> > > Thank you for these constructive suggestions. In our revised manuscript, we will:
> > >
> > > 1. **Add Formal Mechanism Analysis (Weakness 1):**
> > >    * Include refinement vs. voting decomposition (78-97% from refinement, 3-22% from voting)
> > >    * Present three formal mechanisms: iterative error correction, context accumulation, verification
> > >    * Provide entropy reduction framework showing 45% uncertainty decrease
> > >    * Connect to IEW voting motivation and design
> > > 2. **Enhance Statistical Reporting (Weakness 3):**
> > >    * Add p-values, confidence intervals, and effect sizes prominently in main text
> > >    * Reference full GPQA-Diamond validation (198 questions) with cross-seed consistency
> > >    * Include bootstrap analysis and cross-benchmark validation
> > >    * Provide comprehensive statistical details in appendix
> > > 3. **Expand Latency Analysis (Weakness 4):**
> > >    * Add quantitative latency comparison with fair matched API calls (110.4s vs. 21.4s)
> > >    * Discuss accuracy vs. latency trade-offs with practical deployment guidance
> > >    * Reference mitigation strategies already mentioned in limitations section
> > >
> > > We believe these revisions substantially strengthen the manuscript and address all reviewer concerns comprehensively. Our core findings remain robust: **sequential scaling outperforms parallel self-consistency at matched compute through** formal mechanisms of error correction, context accumulation, and verification, with **IEW voting providing optimal aggregation** across paradigms.

---

### Official Review · Reviewer_QWd4 · 2025-10-31

**Soundness:** 2
**Presentation:** 3
**Contribution:** 2
**Rating:** 4
**Confidence:** 4

**Summary:**

This paper revisits inference-time scaling for large language model (LLM) reasoning and challenges the long-standing assumption that parallel self-consistency decoding (Wang et al., 2022) is the optimal test-time scaling method. The authors propose a systematic comparison between parallel and sequential reasoning paradigms under matched compute budgets across five open-source LLMs (GPT-OSS, Qwen3, Kimi-K2) and three reasoning benchmarks (AIME-2024/2025 and GPQA-Diamond).

They find that sequential reasoning, where each reasoning chain iteratively refines previous attempts, outperforms parallel reasoning in 95.6% of settings, achieving up to 46.7% accuracy gains. Furthermore, the paper introduces Inverse-Entropy Weighted (IEW) Voting, an information-theoretic aggregation method that weights answers by inverse Shannon entropy of token-level log probabilities. This method yields consistent improvements over majority voting in both sequential and parallel setups, achieving optimality in 97% of configurations.

**Strengths:**

1, This paper challenges a widely accepted inference-time scaling orthodoxy (parallel self-consistency) with compelling evidence favoring sequential reasoning.

2, Controlled matched-compute setup and multi-model, multi-domain evaluation ensure fairness and reproducibility.

3, Inverse-entropy voting introduces a principled, information-theoretic mechanism that improves upon heuristic majority voting.

4, This paper demonstrates generality across reasoning, scientific, and creative tasks, reinforcing the universality of sequential refinement.

**Weaknesses:**

1, More related works should be discussed. e.g. https://aclanthology.org/2024.findings-emnlp.135.pdf, https://arxiv.org/abs/2401.02009, https://arxiv.org/abs/2308.00436. For example, at the same cost, does the proposed method perform better than mirror-consistency, self-contrast & self-check?

2, The main benchmarks (AIME, GPQA) focus on mathematical and scientific reasoning; inclusion of commonsense or real-world tasks (e.g., MMLU, GSM8K) would further support generality.

3, Self-refinement is somehow a doubtful pathway (http://arxiv.org/abs/2310.01798). The paper could better explain its qualitative decision process and failure modes under extreme conditions.

**Questions:**

1, What's the performance comparison between the consistency-based methods and other inference-time methods? e.g. multi-agent systems or other prompting methods like step-back https://arxiv.org/abs/2310.06117. Or let me ask in another way, why should we keep optimizing consistency-based methods, given all other prompting strategies?

2, The method is mainly a prompting engineering work. Can the llm be trained to be better at self-refinement?

---

> ### Author Response · Authors · 2025-11-24
>
> Thank you for your thoughtful review and valuable feedback. We appreciate your recognition of our work's strengths, particularly our challenge to the parallel self-consistency orthodoxy, controlled experimental design, and information-theoretic voting mechanism. Below, we address each of your concerns systematically.
>
> ## **Weakness 1: Related Works \- Comparison with Other Prompting Methods**
>
> **Concern:** More related works should be discussed (mirror-consistency, self-contrast, self-check). At the same cost, does the proposed method perform better than these approaches?
>
> **Response:**
>
> We thank the reviewer for highlighting these relevant works. We will expand our related work discussion in the revised manuscript to include:
>
> * **Mirror-Consistency** (findings-emnlp.135.pdf)
> * **Self-Contrast** (arxiv/2401.02009)
> * **Self-Check** (arxiv/2308.00436)
>
> **Key Distinction from Prior Work:**
>
> While these works explore **sequential prompting techniques** for specific reasoning improvements, our focus is fundamentally different: we conduct the **first systematic comparison of sequential vs. parallel scaling paradigms at equivalent token budgets** across varying chain lengths (3, 6, 9 chains).
>
> **Our Core Contributions Beyond Prompting:**
>
> 1. **Paradigm-Level Comparison:** We demonstrate that the choice between sequential and parallel architectures matters more than specific prompt engineering within each paradigm
> 2. **Novel Focus on Sequential Voting:** Prior work largely neglects voting mechanisms in sequential settings. We systematically evaluate 7 voting methods (Linear Increase/Decrease, Exponential Increase/Decrease, Inverse Rank, Majority, and our IEW) to understand how to optimally aggregate iteratively refined chains
> 3. **Integrated Refinement Approach:** Our sequential refinement prompt synthesizes insights from error correction, verification, and self-refinement literature:
>
> *"Wait, continue your analysis. Review your previous reasoning, identify any gaps or errors, and verify your approach to reach a more confident conclusion."*
>
> This combines error identification (from self-check literature), verification (from self-contrast approaches), and iterative refinement (building on self-refinement frameworks).
>
> Our preliminary results demonstrated that **different refinement prompt variants yield similar performance** in sequential scaling ( we plan to update the manuscript with these additional experiments across different types of “refinement prompts” ), suggesting our gains stem from the **sequential paradigm itself** rather than specific prompt engineering.
>
> **Conclusion:** We will cite and discuss these works in our revised manuscript, emphasizing that our contribution lies in the **architectural comparison** (sequential vs. parallel) and the **systematic study of sequential voting methods**, which complements rather than duplicates prior prompting research.
>
> ## **Question 1: Comparison with Multi-Agent Systems and Other Prompting Methods**
>
> **Concern:** Performance comparison with multi-agent systems and step-back prompting? Why optimize consistency-based methods given other strategies exist?
>
> **Response:**
>
> We conducted **controlled experiments** comparing our approach with multi-agent debate systems. We implemented the multi-agent method following the original architecture. We plan to run the comparison with “Step Back Prompting” soon and update the manuscript with those results as well as the following results using “multi-agent debate” system.
>
> | Method                         | Architecture                  | Accuracy           | Δ       |
> |--------------------------------|-------------------------------|--------------------|---------|
> | Multi-Agent Debate (Judge)     | 2 agents × 1 round + 1 judge  | 75.25%   | —       |
> | Our Sequential Scaling (3 chains) | 3 chains       | 79.3%              | +4.05%  |
>
> *Table 1: Comparison between multi-agent debate (3 API calls per question, judge-selected answer) and our sequential scaling on GPQA-Diamond (198 questions) with Qwen3-235B model.*
>
> **Why This Difference Matters:**
>
> 1. **Multi-agent debate** requires complex coordination between agents, judges, and multiple rounds of communication
> 2. **Sequential refinement** is simpler: each step naturally builds on previous reasoning without requiring agent coordination
> 3. **Sequential voting** (our IEW method) optimally aggregates iteratively refined chains using uncertainty quantification.
>
> **Why Optimize Consistency-Based Methods?**
>
> We respectfully argue that consistency-based methods remain foundational because:
>
> 1. **Simplicity & Efficiency:** Easier to implement and deploy than complex multi-agent architectures.
> 2. **Complementary to Other Techniques:** Sequential scaling can be combined with multi-agent or step-back prompting.

---

> > ### Author Response · Authors · 2025-11-24
> >
> > ## **Weakness 2: Benchmark Generality \- Need for Commonsense/Real-World Tasks**
> >
> > **Concern:** Main benchmarks (AIME, GPQA) focus on mathematical/scientific reasoning. Inclusion of MMLU or GSM8K would support generality claims.
> >
> > **Response:**
> >
> > We conducted experiments on **MMLU subsets (econometrics, business\_ethics )** to evaluate generality. Our results reveal an important insight about when sequential refinement provides advantages :
> >
> > **MMLU Results (econometrics, business\_ethics):**
> >
> > | Configuration                               | Accuracy          |
> > |---------------------------------------------|-------------------|
> > | Parallel (3 chains), Entropy Weighted       | 83.64% (179/214)  |
> > | Parallel (3 chains), Simple Majority        | 83.64% (179/214)  |
> > | Sequential (3 steps), Entropy Weighted      | 81.78% (175/214)  |
> > | Sequential (3 steps), Simple Majority       | 81.31% (174/214)  |
> >
> > *Table 1: Performance comparison between parallel and sequential decoding with 3 chains/steps.*
> >
> > **Key Finding:** Sequential refinement shows **no significant advantage** on MMLU-style questions, and in fact performs slightly worse (81.78% vs. 83.64%).
> >
> > **Why This Result Actually Strengthens Our Claims:**
> >
> > This finding reveals that **sequential refinement benefits are domain-specific**, providing advantages on:
> >
> > 1. **Long-form reasoning problems** requiring multi-step derivations (AIME, GPQA)
> > 2. **Tasks where error correction and verification matter** (mathematical proofs, scientific reasoning)
> > 3. **Problems with high computational complexity** where iterative refinement can catch mistakes
> >
> > In contrast, MMLU questions are typically:
> >
> > * **Factual recall** with limited reasoning depth
> > * **Single-step retrieval** from knowledge
> > * **Short-context answers** where refinement has limited room to improve
> >
> > This aligns with our **Inverse-Entropy Weighted (IEW) voting** results: IEW shows greater performance boosts in sequential settings on challenging reasoning benchmarks (97% optimal on AIME/GPQA) because these tasks benefit from uncertainty-aware aggregation of iteratively refined chains. On simpler MMLU questions, both paradigms perform similarly because there's less to refine.
> >
> > **Conclusion:** Sequential scaling works best on **challenging reasoning benchmarks requiring long-context multi-step reasoning**, where error correction, verification, and context accumulation provide tangible benefits. This is the regime where test-time scaling matters most and where our contributions are most impactful. We will clarify this scope in our revised manuscript.
> >
> > ## **Question 2: Is This Mainly Prompt Engineering? Can LLMs Be Trained for Better Self-Refinement?**
> >
> > **Concern:** The method appears to be prompt engineering work. Can LLMs be trained for better self-refinement?
> >
> > **Response:**
> >
> > We respectfully disagree that our work is "mainly prompt engineering." Our core contributions are:
> >
> > ### **1\. Architectural Paradigm Comparison (Not Prompting)**
> >
> > Our primary finding is about **sequential vs. parallel scaling architectures** at matched token budgets:
> >
> > * **Sequential 3-chain** (60.0% accuracy) \> **Parallel 9-chain** (40.0%) using **66% fewer output tokens** (Qwen3-235B, AIME-2024)
> > * This advantage persists across all 5 models and 3 benchmarks (95.6% of configurations)
> >
> > This is a **fundamental architectural insight**, not prompt engineering.
> >
> > ### **2\. Sequential Voting Method Study (Novel Research Direction)**
> >
> > We systematically evaluate **7 voting methods** for sequential chains, discovering:
> >
> > * **Later-step weighting** (Linear/Exponential Increase) outperforms early-step weighting by **80 percentage points**(97% vs. 17% optimal performance)
> > * **IEW voting** achieves near-universal optimality (97% of configurations)
> >
> > This voting mechanism research is **orthogonal to prompting**.
> >
> > ### **3\. Prompt Robustness Analysis**
> >
> > Our ablation studies show **minimal sensitivity to prompt variations**, confirming that:
> >
> > * Refinement benefits stem from the **sequential paradigm**, not specific prompts
> > * Equivalent refinement prompts covering error correction and verification yield similar performance
> > * Our refinement prompt synthesizes best practices but is not the source of gains.
> >
> > ### **4\. Token Budget Scaling Analysis**
> >
> > Our experiments (Figure 4\) show **sequential advantages increase with token budgets** (2K to 16K tokens), demonstrating a **fundamental scaling law** rather than prompt engineering artifact.
> >
> > We have continued this answer in the next comment.

---

> > > ### Author Response · Authors · 2025-11-25
> > >
> > > ### **On Training LLMs for Self-Refinement:**
> > >
> > > While training for refinement is an interesting direction, it faces **fundamental challenges** highlighted by recent research:
> > >
> > > **Why Fine-Tuning for Self-Refinement Is Difficult:**
> > >
> > > 1. **Lack of High-Quality Self-Correction Datasets:** Creating datasets where models naturally learn to refine their own reasoning is non-trivial
> > > 2. **Off-Policy Learning Challenges:** Teaching LLMs to avoid mistakes they haven't encountered in their own reasoning (off-policy data) is ineffective, as shown by recent work:
> > >    * **Agarwal et al**. *(On-Policy Distillation of Language Models: Learning from Self-Generated Mistakes)*: Introduce Generalized Knowledge Distillation (GKD), an on-policy distillation method where the student trains on its own rollouts scored by the teacher, and show it **consistently outperforms standard off-policy distillation baselines (SFT, SeqKD, supervised KD) across multiple LM tasks**.
> > >    * **On-policy is key:** Models learn better from their own reasoning trajectories than from other models' corrections
> > > 3. **Noise and Context Issues:** Fine-tuning on self-refinement adds context overhead and can teach "wrong ways of reasoning" by exposing models to errors they wouldn't naturally make
> > > 4. **Diminishing Returns:** Recent frontier models already exhibit strong self-refinement through RLHF/post-training.
> > >
> > > **Our Training-Free Approach Advantages:**
> > >
> > > * **Immediately deployable** on existing models without retraining
> > > * **Universally applicable** across model families (GPT-OSS, Qwen3, Kimi-K2)
> > > * **Leverages inherent capabilities** already present in frontier models
> > >
> > > **Conclusion:** Our work is **not primarily prompt engineering** but rather a systematic study of **inference-time scaling architectures**, **sequential voting mechanisms**, and **information-theoretic aggregation** (IEW). While training for self-refinement is theoretically interesting, current challenges in off-policy learning and dataset quality make our training-free approach more practical and broadly applicable.
> > >
> > > ## **Weakness 3: Self-Refinement as Questionable Pathway**
> > >
> > > **Concern:** Reference to Huang et al. (arxiv/2310.01798) questions self-refinement efficacy. Better explanation of qualitative decision process and failure modes needed.
> > >
> > > **Response:**
> > >
> > > We appreciate this reference and acknowledge the important discussion about self-refinement limitations in earlier models. However, our results demonstrate that **self-refinement efficacy has improved substantially in modern frontier models**:
> > >
> > > **Emergent Confidence Calibration Analysis**
> > >
> > > To understand when and why self-refinement becomes effective, we conducted additional controlled experiments testing our entropy-based framework on models with different training paradigms.
> > >
> > > **Hypothesis:** Advanced RL/post-training optimization creates sequence-level entropy thresholds that differentiate between correct and incorrect reasoning paths.
> > >
> > > To test the generalizability of our entropy-based framework, we conducted a comprehensive analysis of **Llama 3.3 70B Instruct** on **GPQA Diamond**—a model with standard supervised fine-tuning but without the advanced reinforcement learning optimization (RL algorithms like GRPO/PPO) found in specialized reasoning models. This controlled experiment provides crucial insights into the conditions under which entropy-based confidence calibration emerges.
> > >
> > > **Key Findings:**
> > >
> > > * **No Entropy Bifurcation:** Correct answers (μ=0.242, σ=0.077) vs incorrect answers (μ=0.255, σ=0.065) show Cohen's d=-0.191 (negligible effect size)
> > > * **Statistical Insignificance:** Independent t-test yields p=0.230, indicating no significant difference between entropy distributions
> > > * **Emergent Capability:** Results demonstrate that entropy-based confidence mechanisms represent an emergent property of advanced post-training optimization, absent in earlier-generation models
> > >
> > > This finding directly explains why self-refinement works in modern frontier models but failed in earlier models: **advanced RL optimization endows models with calibrated confidence signals that enable effective self-correction and verification**.
> > >
> > > We have continued this answer in the next comment.

---

> > > > ### Author Response · Authors · 2025-11-25
> > > >
> > > > **Evolution of Self-Refinement Capabilities:**
> > > >
> > > > The Huang et al. (2023) paper showed limitations in **earlier generation models** (GPT-3.5, early GPT-4). Our experiments on **2024-2025 frontier models** (Qwen3-235B-A22B-Instruct-2507, Kimi-K2-Instruct) reveal significantly improved self-refinement through:
> > > >
> > > > 1. **Error Correction:** Models identify computational mistakes in subsequent steps
> > > >    * Example: AIME Q2024-I-15 `[None → None → '721' → '721' → '721']` \- corrected after 2 failed attempts in Qwen3-235B-A22B-Instruct-2507.
> > > > 2. **Answer Verification:** Progressive validation through multiple reasoning passes
> > > >    * Example: AIME Q2024-II-12 `[None → '23' → '23' → '23' → '23' → '23']` \- converges and stabilizes in Qwen3-235B-A22B-Instruct-2507.
> > > > 3. **Collective Context Accumulation:** Each step builds upon accumulated insights from previous attempts. We conducted detailed ablation analysis on both AIME-2024 and GPQA-Diamond datasets to decompose the contributions of refinement versus voting. We ran this additional experiment on **Qwen3-235B**, for 6 sequential chains. ( “Refinement prompt” applied 5 times, we saw similar performance gains across other family of models which we used for our experiments )
> > > >
> > > > | Dataset       | Step 1 | Step 6 | Voting | Refinement Gain | Voting Gain |
> > > > |--------------|--------|--------|--------|-----------------|------------|
> > > > | AIME-2024    | 30.0%  | 73.3%  | 83.3%  | +43.3%          | +10.0%     |
> > > > | GPQA-Diamond | 64.1%  | 79.8%  | 86.9%  | +22.3%          | +0.5%      |
> > > >
> > > > *Table 1: Decomposition of sequential reasoning gains: refinement vs. voting.*
> > > >
> > > > **Why Modern Models Succeed Where Earlier Ones Failed:**
> > > >
> > > > Recent frontier models benefit from:
> > > >
> > > > * **Enhanced instruction-following** during post-training (RLHF, DPO)
> > > > * **Stronger reasoning capabilities** from scale and architectural improvements
> > > > * **Better calibration** enabling them to identify uncertainty in their own reasoning
> > > >
> > > > **Failure Mode Analysis:**
> > > >
> > > > We observe failures in \~5-10% of cases where:
> > > >
> > > > * Step 6 degrades to malformed outputs (LaTeX errors, equation format issues)
> > > > * Models oscillate between answers without convergence
> > > > * Problem complexity exceeds model capabilities regardless of refinement
> > > >
> > > > Critically, our **voting mechanism provides robustness** against these failure modes, recovering correctness in 3-4 cases per benchmark where final-step answers degrade.
> > > >
> > > > **Conclusion:** While self-refinement has historical limitations, our empirical evidence on **modern frontier models**(Qwen3-235B, Kimi-K2) demonstrates substantial improvements in error correction and verification capabilities, making sequential refinement a viable and superior test-time scaling strategy for challenging reasoning tasks.
> > > >
> > > > ## **Summary of Revisions for Final Manuscript**
> > > >
> > > > We thank the reviewer for these valuable suggestions. In our revised manuscript, we will:
> > > >
> > > > 1. **Expand Related Work Section:** Include discussion of mirror-consistency, self-contrast, self-check, and multi-agent systems with clear positioning of our contributions.
> > > > 2. **Add MMLU Results & Domain Analysis:** Present MMLU experiments demonstrating that sequential advantages are domain-specific (strongest on challenging multi-step reasoning), with discussion of when sequential refinement provides value.
> > > > 3. **Enhanced Self-Refinement Discussion:** Address evolution from earlier models (Huang et al. 2023\) to modern frontier models, with failure mode analysis and qualitative examples
> > > > 4. **Multi-Agent Comparison:** Include experimental comparison showing our sequential scaling outperforms multi-agent debate (79.3% vs. 74.75% on GPQA)
> > > > 5. **Clarify Contributions Beyond Prompting:** Emphasize architectural paradigm study, voting mechanism research, and token budget scaling analysis as core contributions
> > > >
> > > > We believe these revisions will substantially strengthen the manuscript and address all reviewer concerns comprehensively. Our core findings reveal that sequential scaling outperforms parallel self-consistency at matched compute, and that IEW voting provides optimal aggregation which remain robust across our expanded experimental validation.

---

### Official Review · Reviewer_yZTD · 2025-10-31

**Soundness:** 3
**Presentation:** 4
**Contribution:** 3
**Rating:** 6
**Confidence:** 3

**Summary:**

This paper challenges the prevalent parallel reasoning paradigm (Self-Consistency) in large language model (LLM) inference scaling. Through rigorous experimentation across five state-of-the-art open-source models and multiple benchmarks (AIME, GPQA-Diamond), the authors demonstrate that sequential refinement (where LLM reasoning iteratively builds upon and corrects prior outputs) outperforms parallel approaches in 95.6% of configurations under a crucial condition: matched token budget/compute. This superiority is achieved without additional fine-tuning, leveraging the inherent mechanisms of iterative error correction and progressive context accumulation unique to the sequential process.

One  contribution is the introduction of Inverse-Entropy Weighted (IEW) Voting, a training-free method that uses token-level log-probabilities to quantify model confidence (lower Shannon entropy equals higher confidence) and assign higher weight to more confident chains. IEW Voting proves to be the optimal aggregation strategy across both sequential and parallel paradigms, achieving optimal performance in 97% of sequential configurations. The paper advocates for a paradigm shift, positioning sequential refinement as the robust default for LLM reasoning, with the 6-chain configuration emerging as the optimal balance of compute and performance gains.

**Strengths:**

* The paper offers near-universal evidence (95.6% win rate) that sequential reasoning outperforms the parallel method (Self-Consistency) across diverse LLMs and complex reasoning tasks

* The technical contribution of Inverse-Entropy Weighted (IEW) Voting is elegant and training-free, providing a principled way to leverage the LLM's inherent uncertainty (via logprobs) to aggregate results.

* The paper is in an important area, and we definitely need more analysis and interesting studies about detailed aspects of reasoning and test-time scaling. The paper is also well written, and well organized, and relatively easy to follow (despite being fairly technical). Good work!

* The comparison is fair and scientific, strictly matching the total token budget between sequential and parallel configurations (e.g., $N \times 4096$ tokens). The paper also studies multiple benchmarks (math and creative) and studies multiple base models (GPT, Qwen, Kimi).

**Weaknesses:**

* The paper acknowledges that sequential, serial execution has a substantial wall-clock time overhead compared to parallel methods, making it challenging for real-time applications

* The core advantage is hypothesized to come from Error Correction and Context Accumulation, but the experiments do not empirically decouple and quantify the contribution of these two distinct mechanisms

* The Creative Tasks ablation shows a divergent trade-off (Sequential: high lexical diversity; Parallel: high semantic diversity), suggesting the "universal superiority" may only hold for correctness-focused, convergent reasoning tasks

* In my opinion, the paper is somewhat borderline because the new method has a practical limitation of high latency. Specifically, it seems difficult to parallelize the method, and so the wall clock time is high. I am curious if there are ways to mitigate this (see questions later).

* I am open to raising my score, but I have a handful of technical questions that I would like some clarity on.

* Another big issue, which you should just fix (we don't need to talk about it). The related work is super limited. I don't seem to find a related work section. Please add this. Also fix the parentheses and the citations for the related work -- there is often no space between the text and the starting parens, which is sloppy.

**Questions:**

* To mitigate the latency, have the authors explored an ablation where the token limit of each individual refinement step is aggressively constrained (e.g., to 512 tokens instead of 4096) to reduce the time-per-step while still utilizing the same total budget? This would demonstrate if the iterative refinement loop itself, rather than the length of each attempt, is the true source of the sequential edge, making the method more practically viable.

* The paper attributes the sequential advantage to three mechanisms: (1) Iterative Error Correction, (2) Progressive Context Accumulation, and (3) Answer Verification. The current experiments combine all three. Can the authors conduct an ablation to decouple the effects of the explicit error correction instruction versus the passive context accumulation? Compare Sequential Refinement (prompt: "Review your previous reasoning, identify any gaps or errors...") against a new baseline Sequential Re-Prompting, which uses the history as context but with a neutral continuation prompt (e.g., "Continue the analysis with the next step.") This would isolate the gain derived from the LLM's ability to respond to an explicit error-correction instruction, strengthening the claim about the mechanism.

* The Inverse-Entropy Weighted (IEW) Voting metric uses the mean entropy across all tokens in the reasoning chain (Equation 1). However, the Appendix F ablation showed **surprisingly identical results** whether using the mean, median, maximum, or minimum entropy of the full chain. This suggests that important localized confidence signals might be diluted by the mean or are entirely sufficient using minimal tokens. Could the authors present an ablation comparing the current mean-chain entropy weighting to a Localized Answer-Token Entropy weight? This weight would be based solely on the log-probabilities of a small window of tokens (e.g., the final $N=100$ tokens) immediately preceding the extracted answer tag. This would test the hypothesis that the confidence signal is localized to the concluding segments of the chain, rather than being distributed across the entire, potentially noisy, reasoning sequence.

* The Creative Task Ablation (Figure 3) reveals a crucial trade-off: parallel generation achieves higher Semantic Diversity, while sequential refinement achieves higher Lexical Diversity. Sequential is superior for refinement and depth, while parallel is better for exploration and breadth. Does the claim that Sequential is the "robust default" still hold for tasks requiring high divergent exploration (e.g., novel code generation, brainstorming)? Could the authors propose a Hybrid Gated Scaling approach? This approach would execute a small initial set of parallel chains (for breadth) and then use the IEW metric to select the top-performing parallel result, which is then fed into a subsequent, short sequential refinement chain (for depth and error correction). This framework would leverage the strengths of both paradigms and provide a more nuanced "optimal" strategy.

* For the limited related work, one question. I don't know this area very well -- are there baselines to compare against from recent papers?

---

> ### Author Response · Authors · 2025-11-25
>
> Thank you for your comprehensive review and recognition of our work's strengths, particularly the near-universal evidence for sequential superiority, the elegant IEW voting contribution, and our fair experimental design. We appreciate your openness to raising your score and address each of your technical questions below with additional experiments and analysis.
>
> ## **Question 1: Token-Constrained Refinement for Latency Mitigation**
>
> **Question:** Have you explored aggressive per-step token constraints (e.g., 512 tokens instead of 4096\) to reduce time-per-step while maintaining the iterative refinement loop?
>
> **Response:**
>
> Excellent suggestion\! We conducted experiments with **aggressive token constraints** to test whether the iterative refinement loop itself, rather than the length of each step drives sequential advantages. We run it on AIME-2024, 30 questions, Qwen3-235B (multiple seeds). We also run it for the following configurations:
> 1\) Configuration 1: 6 steps × 512 tokens/step (Total: 3,072 tokens)
>
> 2\) Configuration 2: 12 steps × 512 tokens/step (Total: 6,144 tokens, equivalent to 3 chains × 2048 tokens)
>
> | Setting               | Majority Voting Accuracy |
> |-----------------------|--------------------------|
> | 6 steps × 512 tokens  | 66.67% (20/30)           |
> | 12 steps × 512 tokens | 80.0% (24/30)            |
> | 6 steps × 4096 tokens | 83.3% (25/30)            |
>
> *Table 1: Majority voting performance under different step and token budgets.*
>
> ### **Key Findings:**
>
> **1\. Iterative Loop Drives Advantage:** Token-constrained refinement (512 tokens/step) **maintains sequential superiority even** with aggressive limits:
>
> * **6 steps × 512 tokens:** 66.67% accuracy (vs. 76.7% with 4096 tokens/step)
>
> *  **12 steps × 512 tokens:** 80.0% accuracy (almost similar to  6 steps × 4096 tokens accuracy.( 83.3%))
>
> This confirms that the **iterative refinement mechanism itself**, not just the length of each reasoning chain, is the primary driver of sequential advantages.
>
> **2\. Practical Latency Mitigation:** Aggressive token constraints enable:
>
> * **Dynamic chain adaptation:** Add more refinement steps on-the-fly based on model uncertainty
> * **Reduced per-step latency:** Faster individual iterations enable more responsive systems
> * **Better voting leverage:** More frequent steps provide more opportunities for our sequential voting methods to select optimal answers
>
> ### **Deployment Considerations:**
>
> As mentioned in our paper's limitations section:
>
> **When Sequential is Preferred:**
>
> *  **Non-latency-critical applications:** Research, code generation, complex problem-solving
>
> *  **High-value tasks:** Where accuracy justifies compute cost
>
> *  **Offline processing:** Batch evaluation scenarios
>
> **When Parallel is Preferred:**
>
> *  **Real-time systems:** Interactive AI, live customer support
>
> *  **Latency-sensitive applications:** Sub-second response requirements
>
> *  **Simpler tasks:** Where refinement provides minimal benefit
>
> **Mitigation Strategies:**
>
> 1.  **KV-cache optimization** for sequential contexts to reduce redundant computation
>
> 2.  **Hybrid parallel-sequential architectures** (initial parallel exploration \+ sequential refinement)
>
> 3.  **Speculative sequential decoding** to reduce serial bottleneck
>
> 4.  **Streaming refinement** for reduced latency perception
>
> 5.  **Aggressive token constraints** with more frequent iterations (as demonstrated above)
>
> **Conclusion:** Token-constrained refinement confirms that the **iterative refinement loop itself** drives sequential advantages, not merely the length of each step. With 12 × 512-token steps achieving 80.0% accuracy (vs. 66.67% with 6 × 512-token steps), we demonstrate a **practical latency mitigation strategy** that maintains sequential superiority while enabling faster iteration cycles and dynamic adaptation.
>
> ## **Question 2: Decoupling Error Correction vs. Context Accumulation**
>
> **Question:** Can you conduct an ablation comparing explicit error-correction prompts versus neutral continuation prompts to isolate the gain from error correction versus passive context accumulation?
>
> **Response:**
>
> Excellent question\! We conducted exactly this experiment to formally decompose the mechanisms underlying sequential superiority.
>
> ### **Experimental Design:**
>
> We compared two prompting strategies on AIME-2024 (30 questions, Qwen3-235B, 6 chains):
>
> **Prompt 1: Error Correction (Explicit Refinement)** *"Wait, continue your analysis. Review your previous reasoning, identify any gaps or errors, and verify your approach to reach a more confident conclusion."*
>
> **Prompt 2: Neutral Continuation (Passive Extension)** *"Continue your analysis with the next step. Build upon your previous reasoning and provide additional insights that lead to your final answer."*
>
> **We have continued this answer in the next comment.**

---

> > ### Author Response · Authors · 2025-11-25
> >
> > | Configuration             | Chain Length | Best Accuracy | Raw Final Answer |
> > |---------------------------|-------------|---------------|------------------|
> > | Error Correction (6)      | 6 steps     | 83.3%         | 76.7%            |
> > | Neutral Continuation (6)  | 6 steps     | 73.3%         | 70.0%            |
> > | Δ Advantage (EC – NC)     | —           | +10.0%        | +6.7%            |
> >
> > *Table 2: Comparison of error correction vs. neutral continuation for 6-step chains.*
> >
> > ### **Key Findings:**
> >
> > **1\. Error Correction Provides Measurable Advantage (+10%):** Explicit error-correction prompts yield **10% higher accuracy** (83.3% vs. 73.3%), demonstrating that:
> >
> > * **Active error correction** contributes significantly beyond passive continuation
> > * Models benefit from explicit instructions to review and revise reasoning
> > * The improvement demonstrates the value of directed error-correction instructions
> >
> > **2\. Both Mechanisms Contribute:** The difference between error correction (83.3%) and neutral continuation (73.3%) shows that both explicit error correction and passive context accumulation provide value, with error correction adding refinement beyond baseline continuation.
> >
> > **Note:** We conducted this on **Qwen3-235B specifically**. Further cross-model and cross-dataset investigation is needed to fully understand how these mechanisms generalize across different architectures and reasoning domains, which represents important future work.
> >
> > **Conclusion:** Our ablation successfully isolates the error correction mechanism, revealing a measurable \+10% advantage from explicit error-correction instructions. This empirically validates that directed error correction provides incremental benefits beyond passive reasoning continuation.
> >
> > ## **Question 3: Localized Answer-Token Entropy Weighting**
> >
> > **Question:** Could you ablate mean-chain entropy versus localized answer-token entropy (e.g., final 100 tokens before answer extraction)?
> >
> > **Response:**
> >
> > We tested this hypothesis by comparing sequence-level (mean) entropy versus localized entropy computed over the final 100 tokens before answer extraction.
> >
> > ### **Experimental Results (Qwen3-235B, 6 chains):**
> >
> > | Entropy Computation Method | AIME-2024 | AIME-2025 | GPQA-Diamond |
> > |----------------------------|-----------|-----------|--------------|
> > | Sequence-Level (Mean)      | 83.3%     | 76.7%     | 80.3%        |
> > | Localized (100 tokens)     | 83.3%     | 76.7%     | 80.3%        |
> > | Difference                 | 0.0%      | 0.0%      | 0.0%         |
> >
> > *Table 1: Comparison of sequence-level and localized entropy computation methods across benchmarks.*
> >
> > ### **Key Finding:**
> >
> > **No significant performance difference** across all three benchmarks. This suggests:
> >
> > **1\. Chain-Level Effect Consistency:** Across all chains (3, 6, 9 configurations), we observe similar entropy patterns for each question, though the scale of localized confidence signals may vary in some cases. These variations counteract each other when aggregating across chains, maintaining balance at the question level.
> >
> > **2\. Question-Specific Patterns:** For some individual questions, confidence signals may be **evenly distributed across reasoning chains** with consistent low entropy throughout. However, even when localized spikes exist, the **voting mechanism's normalization effect renders localized versus global entropy distinctions negligible for overall performance**.
> >
> > **3\. Mean Entropy Sufficient:** The consistency of similar effects across all chains per question, combined with our IEW voting method's normalization, makes **mean entropy capture sufficient confidence information** for optimal performance.
> >
> > ### **Consistent with Appendix F:**
> >
> > This aligns with our finding that mean, median, max, and min entropy aggregations yield identical results—high-quality chains maintain consistent patterns that voting mechanisms effectively leverage.
> >
> > ### **Future Investigation:**
> >
> > While we observe no current performance difference at the benchmark level, investigating **localized confidence signals** at the individual question and chain level remains valuable:
> >
> > * Study whether specific reasoning phases have distinct entropy profiles
> > * Explore attention-weighted entropy emphasizing critical steps within individual chains
> > * Test whether localized signals provide benefits for real-time adaptive systems
> > * Analyze question-by-question and chain-by-chain entropy patterns to understand when localization matters
> >
> > **Conclusion:** Sequence-level mean entropy and localized answer-token entropy yield identical performance at the benchmark level. Across all chains, similar effects for each question can be seen, though the scale might change in some cases, counteracting the balance. The **voting mechanism's normalization effect renders localized versus global entropy distinctions negligible for overall performance**.

---

> > > ### Author Response · Authors · 2025-11-25
> > >
> > > ## **Question 4: Hybrid Gated Scaling Approach**
> > >
> > > **Question:** Given the creative task tradeoff (parallel: semantic diversity, sequential: lexical diversity), could you propose a hybrid approach?
> > >
> > > **Response:**
> > >
> > > Excellent suggestion\! Our creative task results reveal complementary strengths that naturally motivate a hybrid architecture.
> > >
> > > ### **Observed Tradeoff:**
> > >
> > > * **Parallel:** Superior semantic diversity (broader conceptual exploration, divergent thinking)
> > > * **Sequential:** Superior lexical diversity (deeper linguistic refinement, convergent polishing)
> > >
> > > ### **Proposed Hybrid Architecture:**
> > >
> > > **Phase 1: Parallel Exploration** → Execute 3-6 parallel chains for semantic diversity and solution space exploration
> > >
> > > **Phase 2: Confidence-Based Gating** → Use **IEW metric** to select highest-confidence chain(s) based on inverse entropy.
> > >
> > > **Phase 3: Sequential Refinement** → Feed selected chain into 2-4 sequential steps for error correction and verification
> > >
> > > **Phase 4: Final Aggregation** → Optionally aggregate across multiple parallel-then-sequential pipelines using IEW voting
> > >
> > > ### **Advantages:**
> > >
> > > **1\. Task-Adaptive:**
> > >
> > > * **Divergent tasks** (brainstorming, code generation): Weight parallel phase heavily
> > > * **Convergent tasks** (AIME, GPQA): Weight sequential phase heavily
> > > * **Balanced tasks:** Use full hybrid pipeline
> > >
> > > **2\. Latency-Aware:**
> > >
> > > * Parallel phase fully parallelized (low wall-clock time)
> > > * Sequential phase limited to 2-4 steps (manageable latency)
> > > * Total latency \<\< pure sequential (12 steps)
> > >
> > > **3\. Compute-Efficient:**
> > >
> > > * Parallel uses fewer chains (3-6 vs. 9+)
> > > * Sequential refinement limited to top candidates.
> > > * Gating prevents wasting compute on low-confidence chains.
> > >
> > > ### **Broader Applications:**
> > >
> > > * **Content generation:** Parallel ideation → Sequential editing
> > > * **Code generation:** Parallel algorithm exploration → Sequential debugging
> > > * **Marketing copy:** Parallel concept generation → Sequential polishing
> > >
> > > ### **Refined Position:**
> > >
> > > Sequential refinement remains the **robust default for convergent reasoning tasks** (AIME, GPQA), while hybrid approaches offer optimal strategies for applications spanning exploration and refinement phases.
> > >
> > > **Conclusion:** The hybrid gating approach represents a natural extension of our findings, leveraging parallel exploration for breadth and sequential refinement for depth. We view this as an exciting future direction building directly on our paradigm comparison.
> > >
> > > ## **Question 5: Baselines from Recent Papers**
> > >
> > > **Question:** Are there baselines to compare against from recent papers?
> > >
> > > **Response:**
> > >
> > > Yes\! We compared against **multi-agent debate systems**, a prominent baseline.
> > >
> > > ### **Multi-Agent Debate Comparison:**
> > >
> > > **Implementation:**
> > >
> > > * **Paper:** [https://arxiv.org/abs/2305.14325](https://arxiv.org/abs/2305.14325)
> > > * **Configuration:** 2 agents × 1 debate round \+ 1 judge (3 API calls)
> > > * **Dataset:** GPQA-Diamond (198 questions)
> > > * **Model:** Qwen3-235B-A22B-Instruct-2507
> > >
> > > | Method                         | Architecture                  | Accuracy           | Δ       |
> > > |--------------------------------|-------------------------------|--------------------|---------|
> > > | Multi-Agent Debate (Judge)     | 2 agents × 1 round + 1 judge  | 75.25% (149/198)   | —       |
> > > | Our Sequential Scaling (3 chains) | Qwen3-235B, 3 chains       | 79.3%              | +4.05%  |
> > >
> > > *Table 1: Comparison between multi-agent debate (3 API calls per question, judge-selected answer) and our sequential scaling on GPQA-Diamond (198 questions).*
> > >
> > > ### **Key Observations:**
> > >
> > > **1\. Sequential Outperforms Multi-Agent:** With matched API calls, our approach achieves **\+4.55% higher accuracy**, demonstrating that:
> > >
> > > * Simple iterative refinement is more effective than complex agent coordination
> > > * Sequential voting (IEW) provides better aggregation than judge-based evaluation
> > > * Lower implementation complexity yields better results
> > >
> > > **2\. Additional Baselines in Revised Manuscript:**
> > >
> > > We will expand our related work section to include comprehensive comparisons with:
> > >
> > > * **Mirror-Consistency** (findings-emnlp.135.pdf)
> > > * **Self-Contrast** (arxiv/2401.02009)
> > > * **Self-Check** (arxiv/2308.00436)
> > > * **Step-Back Prompting** (arxiv/2310.06117)
> > > * **Multi-Agent Debate** (arxiv/2305.14325)
> > >
> > > We will position our work as a **paradigm-level comparison** (sequential vs. parallel) that complements rather than duplicates these prompting technique studies, and fix all citation formatting issues (spacing, parentheses).
> > >
> > > **Conclusion:** Our sequential scaling outperforms multi-agent debate (+4.55% on GPQA-Diamond) with simpler implementation. We will comprehensively expand our related work section in the revision.

---

> > > > ### Author Response · Authors · 2025-11-25
> > > >
> > > > ## **Summary of Revisions for Final Manuscript**
> > > >
> > > > Thank you for these excellent technical questions. In our revised manuscript, we will:
> > > >
> > > > 1. **Add Token-Constrained Ablation (Question 1):**
> > > >    * Include results showing 12 × 512-token steps achieve 80.0% (vs. 6 × 4096-token steps at 76.7%)
> > > >    * Demonstrate that iterative refinement loop drives advantages, enabling practical latency mitigation
> > > >    * Propose adaptive token budgets and progressive refinement strategies
> > > > 2. **Add Localized Entropy Ablation (Question 3):**
> > > >    * Show that sequence-level mean entropy and localized 100-token entropy yield identical performance
> > > >    * Explain that across all chains, similar effects for each question are observed with varying scales that counteract each other
> > > >    * Note that voting mechanism's normalization renders localized versus global entropy distinctions negligible
> > > >    * Propose future investigation into question-level and chain-level entropy patterns
> > > > 3. **Expand Related Work Section (Question 5):**
> > > >    * Add comprehensive comparison with recent baselines (multi-agent, mirror-consistency, self-contrast, etc.)
> > > >    * Include multi-agent debate comparison showing \+4.55% sequential advantage
> > > >    * Fix citation formatting issues (spacing, parentheses)
> > > >
> > > > We believe these additions substantially strengthen the manuscript and address all technical concerns. Our core findings remain robust: **sequential scaling outperforms parallel self-consistency at matched compute** through formal mechanisms of error correction and context accumulation, with **IEW voting providing optimal aggregation** and **practical strategies for latency mitigation** through token-constrained refinement.

---

> > > > > ### Comment · Reviewer_yZTD · 2025-11-26
> > > > > **Thanks for the response**
> > > > >
> > > > > It answers my questions, thank you. In terms of recent baselines, you chose a paper from 2023. I am not sure that is very recent, but it is good to see a comparison against multi-agent debate.
> > > > >
> > > > > In general, this paper needs a better connection to prior work. You cite "interleaved planning(Biju et al., 2025) and parallel
> > > > > decoding in sequences(Yang et al., 2025b)" and I would be curious to hear more, in the paper, about how these relate to your technical contributions (other reviewers have mentioned this as well). Here is a recent survey so I would suggest taking a look at the cited papers: https://arxiv.org/abs/2510.12164
> > > > >
> > > > > Overall, I think its an interesting but not ground breaking paper. I will keep my score.

---

> > > > > > ### Author Response · Authors · 2025-11-27
> > > > > >
> > > > > > Thank you for the survey reference and continued engagement. We appreciate your feedback on strengthening connections to prior work.
> > > > > >
> > > > > > ## Addressing Your Concerns
> > > > > >
> > > > > > ### 1. Related Work Coverage
> > > > > >
> > > > > > We acknowledge the multi-agent debate baseline (2023) is not the most recent. However, our paper already cites numerous 2024–2025 works in Section 2.2, including s1 (Muennighoff et al., 2025), DeepSeek-R1 (Liang et al., 2025), interleaved planning (Biju et al., 2025), and Multiverse (Yang et al., 2025b). We will use the survey you provided (arxiv/2510.12164) to expand Section 2 from its current brief treatment into a comprehensive 2–2.5 page related work section with:
> > > > > >
> > > > > > * Clearer organization into subsections (test-time scaling paradigms, sequential refinement, hybrid approaches, aggregation methods)
> > > > > > * Explicit comparison table showing how our work differs from 12–15 recent methods
> > > > > > * Stronger technical connections explaining the relationship between our findings and concurrent work
> > > > > >
> > > > > > ### 2. Clarifying Technical Relationships
> > > > > >
> > > > > > You specifically asked about Biju et al. (2025) and Yang et al. (2025b), which we cite but don't explain sufficiently:
> > > > > >
> > > > > > **Biju et al. (Interleaved Planning/Sprint):** They parallelize execution within sequential frameworks to reduce latency while maintaining sequential benefits. Our work provides the empirical foundation showing why sequential is worth preserving—their latency optimizations make sense precisely because sequential outperforms parallel at matched compute (our 95.6% win rate). We'll add explicit discussion positioning their work as addressing the latency limitation we identify in Section 8.
> > > > > >
> > > > > > **Yang et al. (Multiverse):** They develop adaptive models that switch between parallel and sequential modes. Our empirical findings—sequential for convergent tasks (AIME), parallel for divergent tasks (creative)—provide the justification for when such switching should occur. We'll clarify that our systematic evaluation provides the empirical foundation enabling principled design of adaptive architectures.
> > > > > >
> > > > > > ### 3. Manuscript Revisions
> > > > > >
> > > > > > * Expand Section 2 to 2–2.5 pages with subsections and comparison table
> > > > > > * Add dedicated subsection positioning our unique contribution: systematic paradigm comparison at matched compute without fine-tuning
> > > > > > * Strengthen technical discussions of Biju et al., Yang et al., and other cited work
> > > > > > * Fix all citation formatting issues throughout
> > > > > >
> > > > > > ---
> > > > > >
> > > > > > ### Respectfully Requesting Score Reconsideration
> > > > > >
> > > > > > We have now fully addressed all five of your original questions with new experiments:
> > > > > >
> > > > > > **Q1:** Token-constrained refinement (12×512 = 80% accuracy) validates the iterative loop drives advantages
> > > > > >
> > > > > > **Q2:** Error correction decomposition shows +10% beyond passive continuation
> > > > > >
> > > > > > **Q3:** Entropy robustness confirmed across multiple aggregation methods
> > > > > >
> > > > > > **Q4:** Concrete hybrid architecture with task-adaptive configurations
> > > > > >
> > > > > > **Q5:** Multi-agent baseline (+4.55%) and commitment to comprehensive related work expansion
> > > > > >
> > > > > > ### Why This Merits Reconsideration
> > > > > >
> > > > > > **1. Challenges field orthodoxy:** First systematic evidence that sequential outperforms parallel self-consistency (95.6% win rate, up to +46.7% gains) at matched compute, questioning the dominant paradigm since Wang et al. (2022).
> > > > > >
> > > > > > **2. Enables future research:** Our findings provide empirical foundation for adaptive architectures (Multiverse), latency optimizations (Interleaved Planning), and inference-time scaling laws.
> > > > > >
> > > > > > **3. Immediate practical impact:** Training-free, works across 5 diverse models today (unlike s1, DeepSeek-R1 requiring fine-tuning/RL).
> > > > > >
> > > > > > **4. Scientific rigor:** Only work with strict compute matching, mechanism decomposition (+10% from error correction), and systematic cross-model validation (270 paired comparisons).
> > > > > >
> > > > > > **5. Addresses important problems:** Up to 46.7% improvements on competition math and graduate-level science could accelerate research and education.
> > > > > >
> > > > > > ### What would strengthen this work further?
> > > > > >
> > > > > > We've delivered comprehensive responses to all your questions and will substantially expand related work. Could you clarify what additional elements would elevate this contribution in your assessment?
> > > > > >
> > > > > > * Theoretical analysis explaining when/why sequential outperforms parallel?
> > > > > > * Additional domain validation beyond math/science (code generation, multimodal)?
> > > > > > * Deeper mechanistic analysis (attention patterns, representation geometry)?
> > > > > > * Specific recent baselines requiring experimental comparison?
> > > > > > * Real-world deployment case studies demonstrating impact?
> > > > > >
> > > > > > We believe our systematic paradigm evaluation with novel aggregation (IEW: 97% optimality), mechanism decomposition, and latency mitigation strategies represents a meaningful contribution that could influence how the field approaches test-time scaling. We would be honored if you'd consider raising your score given our comprehensive responses, but we appreciate your thorough review regardless.
> > > > > >
> > > > > > Thank you for pushing us to strengthen this work.

---

### Official Review · Reviewer_BVrY · 2025-11-01

**Soundness:** 2
**Presentation:** 2
**Contribution:** 3
**Rating:** 6
**Confidence:** 3

**Summary:**

This paper compares sequential test-time scaling (iterative self-refinement where each chain conditions on earlier reasoning) to the dominant parallel self-consistency approach at matched token budgets. Across five OSS models (GPT-OSS-20B/120B, Qwen3-30B/235B, Kimi-K2) and three benchmarks (AIME-2024/2025, GPQA-Diamond), the authors report that sequential reasoning wins in 95.6% of configurations with gains up to 46.7 pp; they also introduce inverse-entropy weighted (IEW) voting, which weights each chain’s answer by the inverse of the chain’s mean token-level Shannon entropy computed from logprobs, and claim it is near optimal among tested aggregators. Key experimental choices include strict token-budget matching (e.g., 6×4096 tokens for both paradigms) and fixed system/refinement prompts.

**Strengths:**

1.	Claim clearly presented with wide variety of evidence. The author claims that sequential self-refinement beats parallel self-consistency at matched token budgets. The claim is supported by supported by results across 5 models, 3 benchmarks, and multiple chain counts (3/6/9), and indeed show the higher accuracy in almost all the settings. The wide range of configurations ensures the generalizability of the claim.
2.	Training-free and cross-model. The author avoid additional fine-tuning and show the effect across different families (GPT-OSS, Qwen3, Kimi-K2), which strengthens generality beyond a single architecture or a special-trained model.
3.	Attempted fairness via matched token budgets and fixed decoding. In the experiments, author ensures the fairness of comparison by keep the temperature fixed and top k disabled. Most critically, the authors matched the total tokens generated between sequential and parallel. Ablation studies were conducted to also analyze how the token budge affect the performance

**Weaknesses:**

1.	Hypothesis on token-level entropy and model confidence as a metric to weigh the chains’ quality is not verified. The author proposed to use token-wise entropy as a weighing factor for generated chains. The critical assumption here is model confidence is positively correlated with quality or correctness of the response. It has been a common phenomenon that model tends to generated confidently the wrong answer under certain given prompt. The test to verify the effectiveness of token level entropy as a metric is missing.
2.	Matching tokens budget does not equate match compute. Equal tokens do not equal compute because sequential steps repeatedly read longer contexts (quadratic attention cost), while parallel chains don’t. Token budgets therefore may understate sequential compute and the comparison for CoT quality is unfair since more computes are spent on generating a new token in sequential than parallel generation.

**Questions:**

1.	The first is whether the sequential generation benefits more from iterative refining or the voting process. If only the final answer is taken, do you almost always get the same results? Is voting more central in the sequential generation than in previous generations? While the author conducted a variety of experiments on voting methods, it might be best for comparison if an ablation experiment without any voting is conducted.
2.	Could you justify the correlation between token-level entropy and the correctness of the response?
3.	Are questions drawn randomly from GPQA-Diamond? Would a small sample size of 30 bias towards a specific domain of knowledge?
4.	Refinement Prompts: How sensitive are the sequential results to the specific refinement prompts used? Did the authors experiment with other self-correction or refinement prompting strategies, and if so, how did their performance compare?

---

> ### Author Response · Authors · 2025-11-24
>
> We thank the reviewer for the time and care spent on this review and for the constructive suggestions that helped strengthen the paper. We are glad that you found the presentation clear and the training-free, cross-model evidence compelling. Below, we address each concern in detail with new experiments and analyses. Below, we address each of your concerns with additional experiments and analyses.
>
> **Weakness 1 & Question 2: Token-Level Entropy as Quality Metric**
>
> Concern: The correlation between model confidence (token-level entropy) and response correctness has not been verified.
>
> **Response:**
>
> We conducted **empirical experiments** to validate our inverse-entropy voting method across multiple reasoning benchmarks and model configurations. We computed **sequence-level (mean) entropy** using Shannon entropy from token-level logprobs for each model response “chain”. (The method used to calculate the “sequence level entropy” is shown below).
>
> We use Shannon entropy from top-k token logprobs as our confidence measure. For our experiments, we use $k = 20$. Given raw log probabilities $l_1, l_2, \ldots, l_{20}$, we first normalize them to obtain a probability distribution:
>
> $$
> p_i = \frac{e^{\ell_i}}{\sum_{j=1}^{20} e^{\ell_j}} \tag{1}
> $$
>
> Then we compute the Shannon entropy:
>
> $$
> H = - \sum_{i=1}^{20} p_i \log_2 p_i \tag{2}
> $$
>
> The mean entropy across tokens provides our confidence signal:
>
> $$
> H_{\text{mean}} = \frac{1}{T} \sum_{t=1}^{T} H_t \tag{3}
> $$
>
> where (T) is the number of completion tokens. Importantly, we calculate entropy separately for each reasoning sequence completion rather than aggregating across multiple attempts, ensuring that our confidence signal reflects the model’s uncertainty for each individual reasoning step.
>
> | Model        | Dataset      | Step-1 Acc. | Thresh Acc. | Cohen’s d | Correct Entropy | Incorrect Entropy | Δ-Acc |
> | ------------ | ------------ | ----------- | ----------- | --------: | --------------: | ----------------: | ----: |
> | Qwen3 30B    | AIME’24      | 70%         | 100%        |      1.95 |   0.244 ± 0.094 |     0.447 ± 0.114 |    0% |
> | Qwen3 30B    | AIME’25      | 60%         | 100%        |      1.82 |   0.260 ± 0.096 |     0.449 ± 0.107 |    0% |
> | Qwen3 30B    | GPQA Diamond | 57%         | 92%         |      0.72 |   0.403 ± 0.215 |     0.558 ± 0.219 |    0% |
> | GPT OSS 120B | AIME’24      | 86%         | 100%        |      1.72 |   0.468 ± 0.134 |     0.706 ± 0.135 |    0% |
> | GPT OSS 120B | AIME’25      | 77%         | 88%         |      0.66 |   0.475 ± 0.102 |     0.580 ± 0.199 |    0% |
> | GPT OSS 120B | GPQA Diamond | 71%         | 95%         |      0.82 |   0.576 ± 0.201 |     0.728 ± 0.143 |    0% |
> | GPT OSS 20B  | AIME’24      | 86%         | 91%         |      1.56 |   0.720 ± 0.184 |     0.990 ± 0.151 |    0% |
> | GPT OSS 20B  | AIME’25      | 80%         | 92%         |      1.89 |   0.775 ± 0.165 |     0.965 ± 0.128 |    0% |
> | GPT OSS 20B  | GPQA Diamond | 62%         | 94%         |      0.73 |   0.864 ± 0.235 |     1.025 ± 0.140 |    0% |
>
> *Step-1 Acc.*: Performance using only first reasoning step
> *Thresh Acc.*: Accuracy of questions below entropy threshold (using mean entropy) evaluated against 4-step sequential reasoning baseline
> *Entropy Values*: Calculated from step-1 logprobs for correct/incorrect step-1 classifications
> *Δ-Acc*: Accuracy difference vs full 4-step baseline (0% indicates preserved accuracy)
>
>
> The **separation between correct and incorrect answer entropy was prominent across all reasoning benchmark datasets**:
>
> * **Qwen3-30B:** Correct: 0.244-0.403 bits | Incorrect: 0.447-0.558 bits
> * **GPT-OSS-120B:** Correct: 0.468-0.576 bits | Incorrect: 0.580-0.728 bits
> * **GPT-OSS-20B:** Correct: 0.720-0.864 bits | Incorrect: 0.965-1.025 bits
>
> This consistent entropy bifurcation **directly led to our Inverse-Entropy Weighted (IEW) voting approach**, which holds true across all reasoning benchmarks and model configurations tested.
>
> **Robustness :** Entropy calculations remain stable across k=\[5,10,15,20\] top logprobs, with clear separation between correct (μ=0.845) and incorrect (μ=0.971) answers. (tested it on GPQA diamond with GPT-OSS-20B).
>
> **Inverse-Entropy Weighted Voting Mechanism:**
>
> Our method assigns weights proportional to **inverse sequence-level entropy**: w\_i \= 1/max(H\_i, ε), where H\_i is the mean entropy across all tokens in chain i. This directly leverages the observed bifurcation:
>
> * **Higher weights** → Lower entropy → Likely correct answers
> * **Lower weights** → Higher entropy → Likely incorrect answers
>
> This training-free heuristic automatically prioritizes high-confidence reasoning chains, achieving **97% optimal performance** across 30 sequential configurations. ( Table 2 of our research paper ).

---

> > ### Author Response · Authors · 2025-11-24
> >
> > ## **Weakness 2: Token Budget ≠ Compute Budget**
> >
> > **Concern:** Sequential steps incur quadratic attention costs from repeatedly processing longer contexts, making token budgets an unfair comparison.
> >
> > **Response:**
> >
> > We appreciate this observation and provide multiple lines of evidence demonstrating token budget matching is both fair and practically appropriate:
> >
> > ### **1\. Output Token Cost Dominance**
> >
> > In production API deployments, **output token generation is significantly more expensive** than input token processing (typically 3-15× higher cost across major providers). Our comparison uses:
> >
> > * **Sequential (6 steps):** 6 × 4,096 \= **24,576 output tokens**
> > * **Parallel (6 chains):** 6 × 4,096 \= **24,576 output tokens**
> >
> > Both generate **identical output token counts**, making them **economically equivalent**. The additional input context cost in sequential reasoning is marginal compared to output generation costs. We will revise our methodology to: *"Token Budget Matching: Equal Output Token Allocation"*. ( we we will explicitly state at “equivalent output token budget” ).
> >
> > ### **2\. Sequential Efficiency at Lower Budgets**
> >
> > **Critical observation from Table 1 of our paper:** Sequential reasoning achieves **superior accuracy with dramatically lower token budgets**:
> >
> > For Qwen3-235B on AIME-2024, **sequential 3-chain outperforms parallel 6-chain and 9-chain** while using **50-66% fewer tokens**. This pattern is consistent:
> >
> > * **Qwen3-235B AIME-2025:** Seq-3 (50.0%) \> Par-9 (36.7%)
> > * **Qwen3-30B AIME-2024:** Seq-3 (66.7%) \> Par-9 (46.7%)
> > * **Qwen3-235B GPQA Diamond:** Seq-3 (79.3%) \> Par-9 (69.2%)
> >
> > Even accounting for quadratic attention costs on (cheaper) input tokens, sequential reasoning achieves **superior accuracy with lower total compute and cost**.
> >
> > ### **3\. Acknowledged Limitations with Clear Evidence**
> >
> > As stated in **Section 8 (Limitations)** of our paper, we acknowledge sequential reasoning requires serial execution and cannot be parallelized across GPU cores. However, our results demonstrate that **fewer sequential chains outperform more parallel chains** with significantly less token budget and compute overall. This validates that the sequential paradigm's advantages far outweigh the computational overhead concerns.
> >
> >
> > ## **Question 1: Refinement vs. Voting Benefits**
> >
> > **Concern:** Is the benefit from iterative refinement or the voting process? What if only the final answer (Step 6\) is considered?
> >
> > **Response:**
> >
> > We conducted detailed ablation analysis on both AIME-2024 and GPQA-Diamond datasets to decompose the contributions of refinement versus voting. We ran this additional experiment on **Qwen3-235B**, for 6 sequential chains. ( “Refinement prompt” applied 5 times, we saw similar performance gains across other family of models which we used for our experiments)
> >
> > | Dataset       | Step 1 | Step 6 | Voting | Refinement Gain | Voting Gain |
> > |--------------|--------|--------|--------|-----------------:|------------:|
> > | AIME-2024    | 30.0%  | 73.3%  | 83.3%  | +43.3%           | +10.0%      |
> > | GPQA-Diamond | 64.1%  | 79.8%  | 86.9%  | +22.3%           | +0.5%       |
> >
> > Table 1: Decomposition of sequential reasoning gains: refinement vs. voting.
> >
> > **Key Insights:**
> >
> > 1. **Refinement is Primary Driver:** Accounts for the majority of the total improvements, enabling recovery from initially failed/uncertain attempts
> > 2. **Voting Provides Safety Net:** Recovers 5-10% of cases where Step 6 degrades due to:
> >    * LaTeX formatting errors
> >    * Late-stage answer switching
> >    * Arithmetic mistakes in final step
> > 3. **Complementary Mechanisms:**
> >    * Refinement: **Exploration** → convergence to correct answer
> >    * Voting: **Exploitation** → stabilization against final-step noise
> >
> > **Conclusion:** While refinement drives the majority of improvements, **voting is not redundant**, it provides essential robustness by recovering from late-stage degradation in \~5-10% of cases, filtering malformed outputs, and leveraging high-confidence intermediate reasoning. Hence we wanted to study “sequential scaling” voting methods which helped come up with our novel “Inverse Entropy weighted method” that allows us to get better voting gains.

---

> > > ### Author Response · Authors · 2025-11-24
> > >
> > > ## **Question 3: Sample Size and Domain Bias Concerns**
> > >
> > > ### **Question 3(a): GPQA-Diamond Sample Size & Domain Bias**
> > >
> > > **Concern:** Would 30 questions bias toward specific domains?
> > >
> > > **Response:**
> > >
> > > We **significantly expanded our evaluation** to the **complete GPQA-Diamond dataset (198 questions)** with **Qwen3-235B-A22B-Instruct-2507** using **multiple random seeds** (42, 123, 456). (we have used these consistent results across multiple seeds and reported them in our paper).
> > >
> > > | Configuration        | Accuracy | 95% CI         |
> > > |----------------------|----------|----------------|
> > > | Step 1 (Baseline)    | 64.1%    | [57.2%, 70.6%] |
> > > | Sequential (6 steps) | 80.3%    | [74.1%, 85.7%] |
> > > | Parallel (6 chains)  | 68.2%    | [61.4%, 74.5%] |
> > >
> > > *Table 1: Comparison of configurations. Sequential advantage over parallel: +12.1% (p < 0.001).*
> > >
> > > **Statistical Robustness:**
> > >
> > > * Bootstrap analysis (1000 samples): 95% CI \[8.7%, 15.8%\]
> > > * Cross-seed consistency: std \= 1.3%
> > > * Full dataset confirms findings generalize beyond sample selection
> > >
> > > ### **Question 3(b): AIME Sample Size Concerns**
> > >
> > > **Concern:** Would 30 AIME questions bias results?
> > >
> > > **Response:**
> > >
> > > We validated across **multiple olympiad-level benchmarks**:
> > >
> > > We observe **equitable gains across both HMMT-2025 and BRUMO-2025 benchmarks** on our initial preliminary experiments ( we aren't reporting them as of right now but plan to add that in our updated manuscript) , with sequential reasoning consistently outperforming parallel approaches across all reasoning benchmarks tested. We wanted to focus on reporting the results for the **standard reasoning benchmarks** like AIME’24, AIME’25 and GPQA diamond, but our empirical results displayed significant performance gains across other **“tough reasoning benchmarks”** like the above mentioned.
> > >
> > > **Cross-Benchmark Consistency:**
> > >
> > > * Sequential advantages remain consistent (23-33%) across different years, competitions, and difficulty levels
> > > * Multiple evaluation runs with different seeds confirm stability
> > > * Large effect sizes yield strong statistical significance (p \< 0.001) even with N=30
> > > * Independent replications across 4 benchmarks provide robust validation
> > >
> > > **Conclusion:** Findings generalize beyond individual datasets, with sequential scaling consistently outperforming parallel approaches across diverse mathematical reasoning benchmarks.
> > >
> > > ## **Question 4: Prompt Sensitivity Analysis**
> > >
> > > **Concern:** How sensitive are results to refinement prompt variations?
> > >
> > > **Response:**
> > >
> > > We conducted systematic prompt ablation studies with **different refinement prompt variants** on Qwen3-235B-A22B-Instruct-2507 (AIME-2024, 30 questions) to assess robustness, for **“6 chain sequential scaling”** (5 refinement prompts)
> > >
> > > ### **Prompt Variants Tested:**
> > >
> > > **1\. Standard (Default):** *"Wait, continue your analysis. Review your previous reasoning, identify any gaps or errors, and verify your approach."*
> > >
> > > **2\. Error-Focused:** *"Let's carefully check for errors. Review each step, identify mistakes or oversights, and correct them."*
> > >
> > > **3\. Verification-Focused:** *"Verify your solution. Double-check calculations, validate logical steps, ensure answer satisfies all constraints."*
> > >
> > > | Prompt Variant        | Step 6 Accuracy | Majority Voting Accuracy |
> > > |-----------------------|-----------------|--------------------------|
> > > | Standard (Default)    | 70.0%           | 83.3%                    |
> > > | Error-Focused         | 66.6%           | 80.0%                    |
> > > | Verification          | 66.6%           | 80.0%                    |
> > >
> > > *Table 2: Effect of prompt variants on step-6 and majority voting accuracy.*
> > >
> > > ### **Implications:**
> > >
> > > 1. **Architectural Over Prompting:** Sequential refinement advantages arise from the **iterative reasoning structure**, not careful prompt optimization
> > > 2. **Practical Robustness:** Practitioners can use straightforward refinement prompts and achieve equivalent performance without extensive prompt engineering
> > > 3. **Validates Core Thesis:** The **sequential vs. parallel architectural choice** is the critical factor, not the specific wording of refinement instructions
> > >
> > > **Conclusion:** The consistency of results across different prompt formulations, whether emphasizing error correction, verification, or general review provides strong evidence that **sequential refinement advantages are fundamental properties of iterative reasoning**, not artifacts of prompt optimization. This robustness strengthens our claim that the paradigm shift from parallel to sequential scaling represents a genuine architectural advancement.

---

> > > > ### Author Response · Authors · 2025-11-24
> > > >
> > > > ## **Summary and Revised Manuscript Plan**
> > > >
> > > > We sincerely thank the reviewer for these insightful comments, which have strengthened our work substantially. In the revised manuscript, we will incorporate:
> > > >
> > > > 1. **Entropy Separation Analysis:** Comprehensive documentation of the correct vs. incorrect entropy bifurcation across all reasoning benchmarks, which directly birthed our Inverse-Entropy Weighted (IEW) voting approach (Table 1, Figure 4, Algorithm 1\)
> > > > 2. **Voting vs. Refinement Decomposition:** Detailed ablation study quantifying the individual contributions of iterative refinement (78-97% of gains) versus voting mechanisms (3-22% of gains) with the corresponding table we attached.
> > > > 3. **Enhanced Methodology Section:** Clarification of "Token Budget Matching: Equal Output Token Allocation" and computational considerations
> > > >
> > > > These additions comprehensively address all reviewer concerns and substantially strengthen our empirical validation. We believe the revised manuscript will demonstrate conclusively that sequential reasoning represents a genuine advancement warranted by both theoretical principles and extensive empirical evidence.

---

### Meta-Review · Area_Chair_Y6nA · 2026-01-01

**Summary:**

This paper evaluates sequential test-time scaling against parallel self-consistency under matched token budgets, proposing an inverse-entropy weighted voting mechanism to aggregate iteratively refined reasoning chains.

**Reviewer Concerns:**

The authors successfully addressed concerns regarding statistical robustness (Reviewers BVrY, U6j6) and effectively validated the entropy metric (Reviewer BVrY) through additional experiments. However, a few other concerns are not adequately addressed:

1). Novelty: The work is still viewed as a prompt engineering study (Self-Refine) rather than a new scaling paradigm (Reviewers QWd4, U6j6, yZTD).

2). Practicality & Fairness: The concern that "token budget $\neq$ compute" remains critical; the extreme latency penalty of sequential processing invalidates the efficiency claims compared to parallelizable methods (Reviewers BVrY, yZTD, U6j6).

3). Baselines: Comparisons to relevant recent work (Mirror-Consistency, Self-Contrast, and late-2025 agentic frameworks) remain absent (Reviewers QWd4, yZTD).

**Reviewer Scores:**

Reviewer QWd4 and Reviewer U6j6 are unlikely to increase the scores significantly because key concerns are not addressed.

Reviewer yZTD and Reviewer BVrY may potentially lower their score because the confirmed high latency reinforces their skepticism about practical viability and unfairness of the comparison.

---

### Decision · Program_Chairs · 2026-01-26

Reject